# AnomalyCoT: A Multi-Scenario Chain-of-Thought Dataset for Multimodal Large Language Models

**Jiaxi Cheng**[1,2*], **Yuliang Xu**[*], **Shoupeng Wang**[*], **Ma Tao**[*], **Yuchen He**[*], **Jinghe Zhang**,
**Sihang Cai**, **Jiawei Zhen**, **Jingyi Jia**, **Yao Wan**, **Yan Xia**, **Zhou Zhao**[1,2†]
[1]Zhejiang University      [2]Shanghai AI Laboratory

## Abstract

Industrial Anomaly Detection (IAD) is an indispensable quality control technology in modern production processes. Recently, on account of the outstanding visual comprehension and cross-domain knowledge transfer capabilities of Multimodal Large Language Models (MLLMs), existing studies have explored the application of MLLMs in the IAD domain and established some multimodal IAD datasets. However, although the latest datasets contain various fundamental IAD tasks, they formulate tasks in a general question-and-answer format lacking a rigorous reasoning process, and they are relatively limited in the diversity of scenarios, which restricts their reliability in practical applications. In this paper, we propose AnomalyCoT, a multimodal Chain-of-Thought (CoT) dataset for multi-scenario IAD tasks. It consists of 37,565 IAD samples with the CoT data and is defined by challenging composite IAD tasks. Meanwhile, the CoT data for each sample provides precise coordinates of anomaly regions, thereby improving visual comprehension of defects across different types. AnomalyCoT is constructed through a systematic pipeline and involves multiple manual operations. Based on AnomalyCoT, we conducted a comprehensive evaluation of various mainstream MLLMs and fine-tuned representative models in different ways. The final results show that Gemini-2.0-flash achieved the best performance in the direct evaluation with an accuracy rate of 59.6%, while Llama 3.2-Vision achieves the best performance after LoRA fine-tuning with an accuracy rate of 94.0%. Among all the fine-tuned models, the average accuracy improvement reaches 36.5%, demonstrating the potential of integrating CoT datasets in future applications within the IAD field. The source code and data are available at `https://github.com/Zhaolutuan/AnomalyCoT`.

## 1   Introduction

Industrial Anomaly Detection (IAD) has emerged as a crucial requirement in the modern industrial production processes. IAD facilitates the real-time monitoring and identification of product anomalies using automated techniques, thereby enabling timely intervention to prevent the propagation of errors and maintain product quality. Existing IAD methods exhibit high real-time performance and accuracy, effectively mitigating economic losses in industrial production and facilitating the transition towards unmanned operations [1, 2]. With the rise of Multimodal Large Language Models (MLLMs) , *i.e.* GPT-4 [3] and Gemini [4], which can undertake various human tasks, recent research has also attempted to apply MLLMs to IAD tasks. AnomalyGPT [5], as the first method to use MLLMs to solve IAD tasks, overcomes the limitation of most past IAD methods that require manual threshold setting for anomaly detection and is directly fine-tuned on IAD datasets. Furthermore, some studies utilize the joint modeling capacity of MLLMs and combine instruction fine-tuning methods [6, 7]

---

[*]Equal contribution.

[†]Corresponding Author.

39th Conference on Neural Information Processing Systems (NeurIPS 2025) Track on Datasets and Benchmarks.

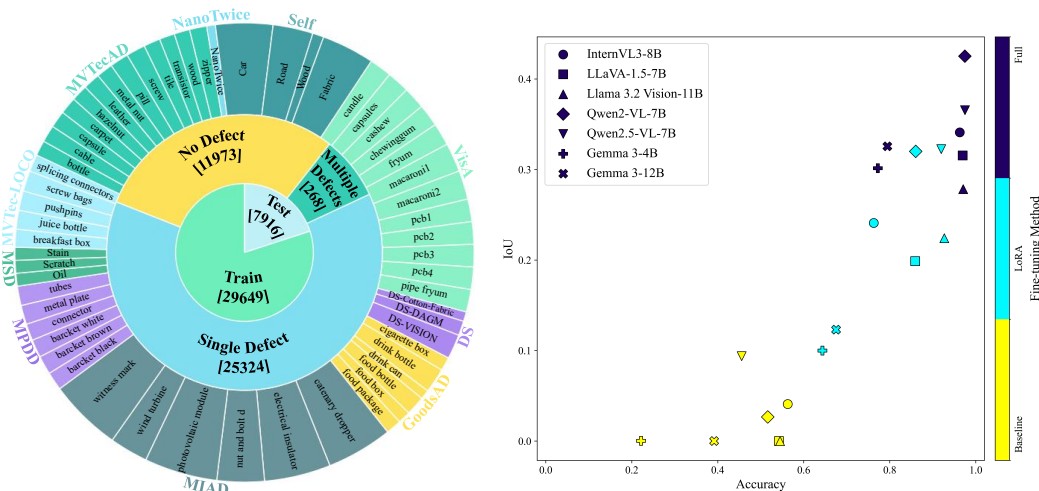

Figure 1: Left: The outermost layer shows the datasets that make up AnomalyCoT and the corresponding objects, the middle layer shows the number of different defects, and the innermost layer shows the division of the training set and the test set. Right: The performances of different MLLMs, including pre-trained, LoRA fine-tuned, and full fine-tuned models, are shown in this figure.

to solve IAD tasks using the visual question-answering paradigm, demonstrating the flexibility of MLLMs in task adaptation [8, 9].

In recent years, significant progress has been achieved in applying MLLMs to Industrial Anomaly Detection (IAD) tasks, concurrent with the development of a series of high-quality IAD datasets. MVTec AD [10] is one of the most influential datasets in the IAD field, not only containing over 5,300 high-resolution images of multiple objects exhibiting diverse defect types, but also providing pixel-precise annotations for all anomalies. This facilitates its application in both binary classification and anomaly detection tasks, thereby significantly advancing the development of IAD methods. MIAD [11] is a brand-new large-scale outdoor IAD dataset, containing over 100K high-resolution images and covering various types of structural and logical anomalies, providing strong support for the application of IAD tasks in **outdoor scenarios** and addressing the previous scarcity of outdoor anomaly samples. Other studies, such as MMAD [12], have proposed the first benchmark for evaluating the comprehensive performance of MLLM in IAD tasks. By designing multiple key subtasks of MLLM in anomaly detection and collecting public IAD datasets, a multimodal benchmark dataset in the form of Q/A for IAD was constructed. This benchmark specifically addresses the limited availability of dedicated resources for applying MLLMs within the industrial domain.

While the aforementioned datasets have made significant contributions to IAD research and application, they are subject to certain limitations. Firstly, MVTec AD and MIAD concentrate on manufacturing and maintenance inspection scenarios, respectively. This domain-specific focus and limited data diversity restrict their generalizability and direct applicability to broader real-world environments. Secondly, MMAD provides multi-angle IAD QA and general anomaly localization information, enabling the determination of sample status (normal/anomalous). Nevertheless, it does not offer a systematic analysis explaining the reasons for the observed normal or anomalous states. Additionally, since MMAD only employs a simple question-and-answer format, which lacks interpretability in the decision-making process and logical reasoning in intuitive representation, it becomes difficult to determine its reliability in practical applications. Overall, existing IAD datasets exhibit significant potential for expansion and require further refinement to enhance their applicability to real-world scenarios. Therefore, developing a systematic Chain-of-Thought (CoT) dataset is of particular importance, as it can facilitate the deployment of MLLM in IAD tasks within complex, dynamic environments.

In this paper, we introduce **AnomalyCoT**, the first multimodal CoT dataset for IAD tasks. To address the limitations of the current IAD datasets and facilitate the application of MLLM in real-world IAD tasks, our dataset offers three significant advantages. (i) The first advantage of the dataset lies in its **extensive coverage of IAD scenarios**. As illustrated in the left panel of Figure 1, we achieve substantial growth in both data scale and scenario diversity by systematically integrating and

performing unified preprocessing on a wide range of publicly available IAD datasets, all anomaly-related CoT annotations were newly constructed and manually verified by us. Moreover, we uniformly re-labeled anomalous regions instead of reusing original annotations to ensure consistency. In addition, our newly collected Self dataset contributes 8,466 images, which further enlarges the dataset scale; (ii) Another advantage of the dataset is that it unifies the two core tasks of IAD, namely anomaly discrimination and defect classification, to define an **end-to-end detection task**. Meanwhile, it includes the analysis process of a large number of normal or anomalous samples. We design the dataset with a question-reasoning-answer structure and provide no defect options in each sample's question options, enabling a complete analysis of normal or anomalous samples based on anomaly judgment; (iii) Finally, the most significant advantage of our dataset is its **interpretable reasoning process**, which provides precise coordinates of anomalous regions for anomalous samples to assist in analysis. To effectively address the ambiguity of decision-making basis in the simple question-and-answer form, we design and generate rigorous reasoning processes by combining accurate visual information of anomalous regions and the rich knowledge contained in large models, ensuring the reliability of the dataset in practical applications.

To fully manifest the aforementioned advantages of the dataset and exert the comprehensive capabilities of MLLM in addressing IAD tasks, we constructed a novel pipeline. Firstly, we collected the public datasets dedicated to studying different IAD tasks, forming a comprehensive IAD scenario. Subsequently, we designed a composite IAD task encompassing no defect options and distractors to achieve the synergetic completion of anomaly discrimination and defect classification. By **manually annotating** a large amount of data and precisely locating anomalous regions in the form of coordinate pairs, we have also designed a general semantic prompt embedded with rich visual information. Combined with QwenVL-Max, we generate a clear and logically rigorous reasoning process to establish a complete CoT data. Specifically, we combined the coordinates of the anomalous area, the defect type, and a reasoning template as prompts to guide QwenVL-Max in generating initial CoT. Additionally, we carried out rule-based filtering and iterative updates on the CoT data. On this basis, all the data were manually verified to ensure the accuracy and rationality of the CoT. Our approach is consistent with Diff-Prompt[13] in terms of prompt/CoT construction, which uses a diffusion model to generate fine-grained prompt representations, thereby improving the performance of complex tasks. The MultiModal Conditional Retrieval[14] proposed in KDD 2025 studied how to perform highly controllable reasoning under multimodal conditions, which has reference significance for the design of cross-modal prompts in this method. Finally, we collected 37,565 samples from 59 types of scenarios in 13 public datasets.

In the experiment, we first conducted a comprehensive evaluation of various mainstream MLLMs using AnomalyCoT, such as the open-source models Qwen2.5-VL [15] and InternVL3 [16], commercial models GPT-4o [17] and Gemini-2.0-flash, and the IAD model AnomalyGPT. We also perform different types of fine-tuning training on representative MLLMs like Qwen2.5-VL and choose Intersection over Union (IoU [18]) as the inference evaluation metric due to the post-training inference processes mostly including the coordinates of anomaly regions. The experimental results indicate that by adopting different fine-tuning methods on the major MLLMs, both the accuracy and IoU of the models are significantly improved compared to the case of direct evaluation, as shown in the right panel of Figure 1. This reflects the effectiveness of our dataset in the IAD task. Additionally, we have conducted thorough ablation experiments under various settings. Specifically, we compared models trained without CoT annotations and without coordinate information to highlight the necessity of both components. Overall, our dataset has played a significant role in challenging major IAD tasks and demonstrates great potential for application in future IAD tasks with high-precision requirements.

Our contributions are summarized as follows:

- We construct AnomalyCoT, a new dataset for testing the comprehensive reasoning capabilities of mainstream MLLMs in the IAD task. To the best of our knowledge, our proposed dataset is **the first multimodal CoT dataset in the IAD task**. This dataset sets new requirements for the application logic of MLLMs in the industrial field.
- We have significantly expanded the scenarios of the IAD task and introduced a new pipeline for generating accurate and rigorous reasoning processes for anomaly detection tasks.
- We adopt structured CoT data and conduct fine-tuning experiments on representative MLLMs to comprehensively evaluate the performance of MLLMs on AnomalyCoT and achieve normative analysis of MLLMs in specific tasks.

## 2 Related Work

### 2.1 Industrial Anomaly Detection

Industrial anomaly detection plays a crucial role in ensuring product quality and production safety, which makes the development of more generalizable and explainable methods especially valuable. Traditional IAD research have primarily focused on locating and classifying defects in novel environments. Common IAD methods include reconstruction-based methods [19, 20], synthesis-based methods [21], and feature embedding-based methods [22, 23, 24]. Reconstruction-based methods learn the reconstruction capability of normal samples and calculate the reconstruction error to detect anomalies. Synthesis-based methods employ data augmentation techniques to synthesize anomalous samples, transforming the original problem into a binary classification task distinguishing between normal and synthesized anomalous instances. Feature embedding-based methods model the feature distribution of normal samples and quantify the feature deviation of test samples. These methods typically require learning the distribution of a large number of samples of existing categories, making it difficult to learn new category instances in dynamic environments. Recent research has predominantly focused on performing IAD tasks with few-shot learning. Several studies integrated visual language models like CLIP, such as InCTRL [25], which employs a few positive samples as contextual prompts to learn the residual between test samples and prompts based on CLIP. AnomalyCLIP [26] learns object-agnostic textual prompts to capture different features within samples, focusing on anomalous regions. However, these models overemphasize predefined anomaly concepts, resulting in limited generalization in new scenarios. Given the flexibility of MLLMs in handling complex visual and textual inputs, addressing the aforementioned limitations becomes possible.

Some recent studies have focused on applying MLLMs to IAD tasks and yielding promising outcomes. MMAD has established an MLLM benchmark test encompassing seven key sub-tasks of IAD, conducting a comprehensive evaluation of various mainstream MLLMs. Other studies directly fine-tune MLLMs using public IAD datasets, such as AnomalyGPT and FabGPT [27], but the performance of such models is often influenced by their expert models. Moreover, models like AnomalyGPT not only are susceptible to overfitting due to the limited scale of IAD data but also have rough anomaly localization and lack rigorous reasoning processes to substantiate detection results, hindering their applicability in real-world scenarios. Consequently, proposing the first multimodal chain-of-thought dataset based on IAD tasks is of great significance. Traditional IAD methods such as SPADE, PaDiM, and PatchCore rely on pixel reconstruction or feature embedding distance, which makes them effective in detecting surface or texture anomalies under controlled settings. Recently, FUSION has adopted a text-guided unified visual encoding and recursive alignment decoding mechanism to achieve tighter cross-modal fusion. However, they struggle to address logical anomalies and often require separate models for each object type. In contrast, VLLM-based approaches leverage cross-modal reasoning and can simultaneously handle anomaly detection and defect classification, demonstrating stronger adaptability in multi-scenario industrial environments.

### 2.2 Multimodal Large Language Model

MLLM integrates multiple modalities, including vision, into LLMs to form large models endowed with visual understanding capabilities. The cross-modal information input to MLLM is initially mapped to the text modality and subsequently processed by LLM. This modeling paradigm has demonstrated robust performance in a series of visual tasks. Early research such as BLIP2 [28] and Flamingo [29] adopted the frozen visual encoder paradigm, achieving end-to-end visual question answering capabilities through alignment with LLM. Subsequently, the LLaVA series [30] and MiniGPT-4 [31] introduced visual instruction fine-tuning methods, significantly enhancing the ability to follow intricate instructions. Models like Qwen-VL series [32], VisionLLM [33], and KOSMOS2 [34] integrate regional visual features to bolster the visual foundation capabilities of MLLM, enabling it to perform tasks such as regional semantic localization. InternVL [35] expands the visual encoder and optimizes cross-modal attention to align with LLM semantics. Additionally, Gemini exhibits cross-domain general visual understanding capabilities through large-scale multimodal pre-training. Models such as DeepSeek-VL2 [36] and Uni-MoE [37] employ a mixture-of-experts architecture to enhance multimodal understanding capabilities. Recently, in the field of multimodal fusion, AudioVSR[38] improves performance by injecting audio information into visual models and OmniCam [39] integrates diffusion-based video generation with large language models through camera control explored unifying multimodal generation and reasoning under controllable

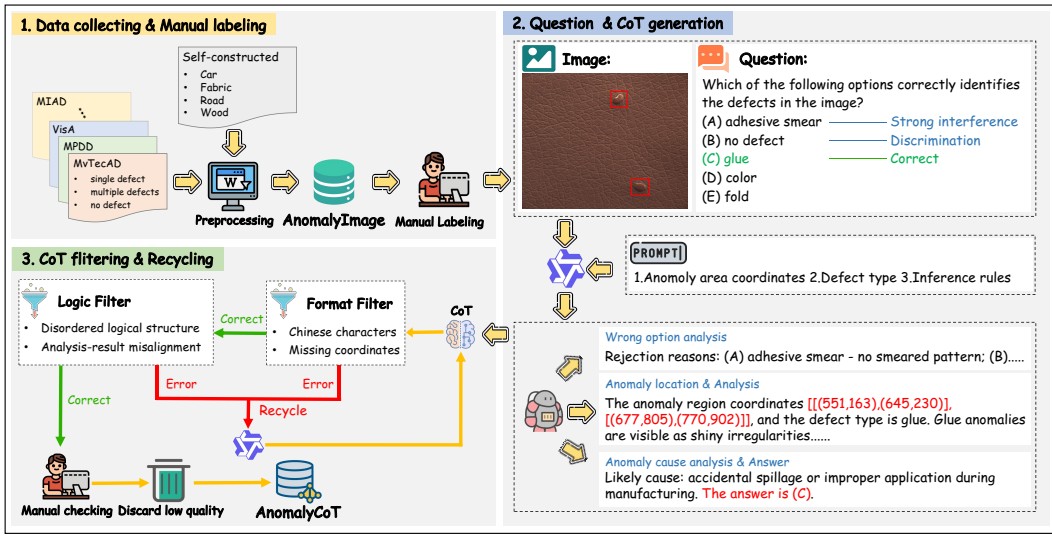

Figure 2: Data generation process of AnomalyCoT. The construction of AnomalyCoT mainly conserns three parts: the collection of image data and manual labeling, the generation of questions and CoT, the filtering and recycling of CoT data.

frameworks. However, existing MLLMs still struggle to reason as naturally as humans, which limits their effectiveness in real-world complex environments. Therefore, fine-tuning training on a large amount of chain-of-thought data from multiple scenarios on mainstream MLLMs can enable them to acquire good reasoning abilities to handle challenging visual tasks.

# 3 Dataset

## 3.1 Data Collection

Anomaly detection tasks in real-world settings are typically complex and variable, and various types of defects can occur in different products. Therefore, the datasets we construct are required to cover multiple scenarios and defect types of anomaly detection. We initially acquired and sampled from nine publicly available IAD datasets. To further expand the scenarios, we also manually gathered data from four scenarios applicable to IAD tasks to form several independent datasets and eliminated low-quality data through manual examination. MVTec AD is one of the most renowned IAD datasets, containing high-resolution images of various objects and corresponding to different types of defects. The VisA [40] dataset further enhances the complexity of scenarios and the data scale, including different instances of the same type of objects. GoodsAD [41] offers the possibility for intelligent applications of anomaly detection by establishing the first IAD commodity dataset. MVTec LOCO AD [42] is dedicated to exploring logical anomalies and constructs a novel dataset covering both structural and logical anomalies. We also adopted Defect Spectrum [43], which effectively optimizes based on multiple key IAD datasets.

In industrial fields with less attention, MPDD [44] and NanoTwice [45] are respectively designed for anomaly detection tasks in the production processes of painted metal parts and nanofiber materials. MSD [46] is a potential dataset for surface defect detection of smartphone screens. MIAD is the largest component of the dataset in this paper, containing over 100K high-resolution outdoor IAD images. The large-scale, high-quality data make a significant contribution to the application of IAD tasks in practical environments. Additionally, we collected IAD data related to wood, fabric, car, and road cracks, screened them for high-quality samples, and then manually classified the unlabeled data to build four independent datasets. These independent datasets fill the voids in certain scenarios of the current task. We integrated the aforementioned processed datasets, precisely located all anomalous regions, and classified the defects within them through manual annotation, then verified the correctness of the labels through manual inspection, and finally manually corrected the incorrect labels. A total of 10 individuals participated in this labeling process, which accumulated to 230 hours. Although the dataset integrates multiple public sources, we recognize that differences in source data distributions may introduce potential biases toward certain industrial sectors. To mitigate

this, we balanced the sampling ratios across sub-datasets and further evaluated the fine-tuned model on held-out datasets (MVTec AD, MVTec LOCO AD, VisA, and GoodsAD), where the model still maintained strong performance. These results suggest that the learned representations are robust to distributional shifts.

## 3.2 Question Construction

In the actual production process, operators typically focus on whether there are anomalous situations in the products and the possible types of defects. When necessary, they also need to understand the scale and specific location information of the anomalies to analyze the causes and prevent subsequent influences. Thus, we designed a key task integrating anomaly discrimination and defect classification, generating a reasoning process that combines anomalous region information with cause analysis grounded in the framework of this task. We presented the task in the form of single-choice questions to evaluate the output of MLLM. Previous work [47, 48] has also demonstrated the rationality of this approach. To effectively implement this complex task and avoid the inherent biases in MLLM, we first defined the no defect option and manually designed strong distractors with high semantic similarity to the corresponding answer items of the samples. Subsequently, we manually constructed defect type libraries for different scenarios and randomly selected other defect types from them as supplementary options, ultimately forming a difficult task with five options.

## 3.3 Data Generation

The existing public IAD datasets lack the reasoning process from task to result, which makes it difficult for MLLMs to conduct rigorous evaluations of IAD tasks directly. Therefore, we have constructed a systematic pipeline to generate complete CoT for IAD samples. Our process utilizes the outstanding visual perception and text reasoning capabilities of QwenVL-Max, combined with manual annotation, semantic prompts, and rule-based filtering mechanisms, as depicted in Figure 2. Firstly, we design challenging questions for IAD samples. Since most MLLMs provide rough anomaly localization in text descriptions when performing IAD tasks and have difficulty forming reasoning processes that conform to the thinking patterns of professionals, we further design comprehensive task prompts. The task prompts for samples not only include precise visual prompts such as the coordinates of the annotated anomalous regions but also semantic prompts like defect type labels and standardized reasoning guidance. By learning the rich prompt content, QwenVL-Max generates a rigorous reasoning process consisting of important components such as the analysis of incorrect options, analysis based on anomaly localization, analysis of the causes of anomalies, and the answer. This reasoning process is then combined with the task to form complete CoT data, effectively simulating the cognitive patterns humans employ when addressing analogous problems.

To ensure the accuracy and rationality of the generated CoT, we have developed a two-step filtering approach. In the first stage, since QwenVL-Max may cause problems such as Chinese characters, the CoT data undergoes Chinese character detection and missing coordinate detection to separate data with obvious format errors. Subsequently, the data undergoes logical error filtering in the second step. We integrate LLMs with superior natural language comprehension to filter data samples exhibiting both analysis-answer misalignment and disordered logical architectures (such as reversed cause and effect), with representative examples demonstrated in Appendix E. We also re-collected samples corresponding to the erroneous data to minimize data loss. Finally, all the data that have passed through the two-step filtering are manually inspected, and low-quality data are discarded to form our dataset AnomalyCoT. This inspection process involved 5 individuals and took a cumulative 10 hours. Figure 3 illustrates the four CoT data types in AnomalyCoT. While our dataset construction currently involves a certain amount of manual annotation for CoT verification, our experiments demonstrate that models fine-tuned on the existing dataset can generalize to unseen domains (e.g. MVTec AD, MVTec LOCO AD, VisA, GoodsAD) without requiring additional annotations. This suggests that the dataset can be feasibly extended to new domains with reduced annotation overhead in the future.

# 4 Experiments

## 4.1 Settings

To understand the effectiveness of AnomalyCoT, we conducted comprehensive evaluation on a series of MLLMs. For each question, the model is required to choose the correct option and identify

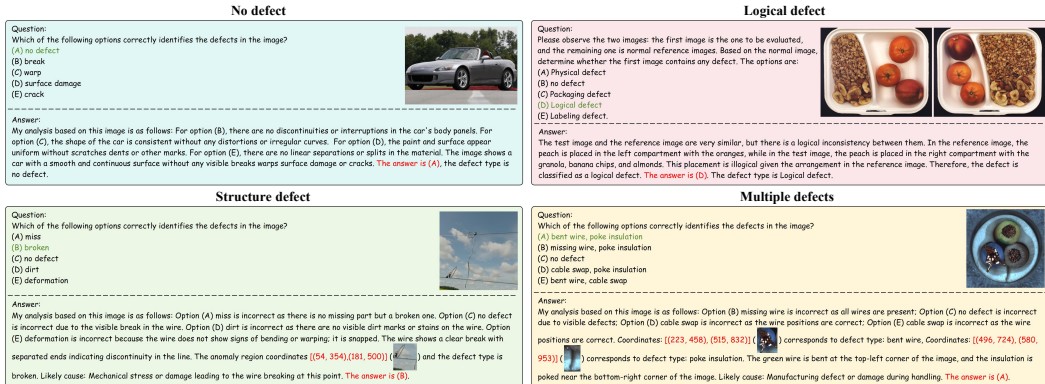

Figure 3: Samples of different defects. In AnomalyCoT, there are samples with different defects including normal image without any defect, single defect, multiple defects and logical defect.

the coordinates of the anomalous region, which not only reflect the model's ability to classify and locate anomalies, but also avoid the confabulations[49] introduced by LLM. According to these, we report two aspects of metrics: accuracy and different types of IoU. The test set was constructed by proportionally sampling from each sub-dataset. Larger-scale sub-datasets were assigned a relatively lower ratio of test samples, while smaller sub-datasets received proportionally higher allocations, thereby maintaining an overall balanced train-test split. The scene distribution and defect type ratios are summarized in Table 4 of Appendix A.

**Baselines.** Though many MLLMs have strong capability in vision-text comprehension, they may not generalize well in IAD tasks. Therefore, to understand the basic capabilities of MLLMs in this specific domain, we conduct evaluation on both commerical and open-source models. For commercial models, we tested GPT-4o and Gemini-2.0-flash. For open-source models, we tested and adapted AnomalyGPT, InternVL3, LLaVA-1.5 [50], Llama 3.2-Vision[3], Qwen2-VL [51], Qwen2.5-VL [15] and Gemma3 [52].

**Comparison.** We further compared AnomalyCoT with representative traditional IAD methods, including SPADE, PaDiM, and PatchCore, on the MVTec AD dataset. Since these methods mainly focus on anomaly detection and cannot perform fine-grained defect classification, Image-AUC was adopted as their primary evaluation metric, while MLLM-based approaches were evaluated using classification accuracy. Results show that traditional methods achieve strong anomaly detection ability but are limited in handling classification tasks, whereas fine-tuning on AnomalyCoT enables MLLMs to achieve competitive performance across both anomaly detection and defect classification. In addition, we compared against recently proposed anomaly generation methods that support classification, namely AnomalyDiffusion[53] and DualAnoDiff [54]. As shown in Table 8, these methods achieve accuracy below 80% on MVTec AD, while our fine-tuned Llama 3.2-Vision model significantly outperforms them, reaching 87.65%. This highlights the reliability of our approach in practical scenarios and its advantage in addressing complex multimodal IAD tasks. For MLLM-based methods, we adopt accuracy as the primary evaluation metric since the models can directly output defect categories in addition to anomaly detection. For traditional IAD methods, which mainly detect the presence of anomalies without classification, we follow prior work and report Image-AUC. While these differences limit direct metric-level comparisons, together they provide a comprehensive view of detection versus classification capabilities.

**Fine-tuning.** We conduct both LoRA and full fine-tuning for selected models on AnomalyCoT. The tuning experiments are relied on LLaMA-Factory [55] and the samples are organized by standardized sharegpt format as shown in Appendix C. We train both models with batch size 32 by 3 epochs and more detailed configuration is recorded in Appendix D. In addition, the system prompt used for questioning is shown in Appendix B.

---

[3] https://github.com/worldart/meta-llama_llama-models/tree/main/models/llama3_2

Table 1: Overall evaluation results.

| Model | Scale | Type | Accuracy | IoU | GIoU | DIoU |
|---|---|---|---|---|---|---|
| GPT-4o | - | Pre-trained | 49.19% | 0.0364 | -0.2529 | -0.102 |
| Gemini-2.0-flash | - | Pre-trained | 59.64% | 0.0413 | -0.1923 | -0.057 |
| AnomalyGPT | 7B | Pre-trained | 19.17% | 0.0000 | -0.0414 | -0.0306 |
| InternVL 3 | 8B | Pre-trained | 57.33% | 0.0405 | -0.1159 | -0.0272 |
| | | LoRA | 91.27%(+33.94%) | 0.2667(+0.2262) | 0.0153(+0.1312) | 0.1747(+0.2019) |
| | | Full | 94.48%(+37.15%) | 0.4045(+0.3640) | **0.2154**(+0.3313) | 0.3532(+0.3804) |
| LLaVA-1.5 | 7B | Pre-trained | 28.84% | 0.0000 | 0.0000 | 0.0000 |
| | | LoRA | 59.12%(+30.28%) | 0.2123(+0.2123) | -0.0772(+-0.0772) | 0.0999(+0.0999) |
| | | Full | 96.94%(+68.10%) | 0.3155(+0.3155) | 0.1139(+0.1139) | 0.2540(+0.2540) |
| Llama 3.2-Vision | 11B | Pre-trained | 55.07% | 0.0001 | -0.0110 | -0.0054 |
| | | LoRA | 94.02%(+38.95%) | 0.2483(+0.2482) | 0.0194(+0.0304) | 0.1635(+0.1689) |
| | | Full | 97.07%(+42.00%) | 0.2784(+0.2783) | 0.0566(+0.0676) | 0.2059(+0.2113) |
| Qwen2-VL | 7B | Pre-trained | 51.61% | 0.0268 | -0.0409 | -0.0023 |
| | | LoRA | 86.92%(+35.31%) | 0.3201(+0.2933) | 0.1063(+0.1472) | 0.2496(+0.2519) |
| | | Full | 97.44%(+45.83%) | **0.4252**(+0.3984) | 0.2076(+0.2485) | **0.3642**(+0.3665) |
| Qwen2.5-VL | 7B | Pre-trained | 45.53% | 0.0939 | 0.0751 | 0.0864 |
| | | LoRA | 91.94%(+46.41%) | 0.3225(+0.2286) | 0.1069(+0.0318) | 0.2562(+0.1698) |
| | | Full | **97.46%**(+51.93%) | 0.3655(+0.2716) | 0.1462(+0.0711) | 0.3029(+0.2165) |
| Gemma 3 | 4B | Pre-trained | 22.15% | 0.0003 | -0.0238 | -0.0141 |
| | | LoRA | 64.33%(+42.18%) | 0.1000(+0.0997) | -0.2447(+-0.2209) | -0.0316(+-0.0175) |
| | | Full | 77.23%(+55.08%) | 0.3015(+0.3012) | 0.1329(+0.1567) | 0.2556(+0.2697) |
| Gemma 3 | 12B | Pre-trained | 39.12% | 0.0002 | -0.0149 | -0.0085 |
| | | LoRA | 67.48%(+28.36%) | 0.1232(+0.1230) | -0.1434(+-0.1285) | 0.0142(+0.0227) |
| | | Full | 79.42%(+40.30%) | 0.3257(+0.3255) | 0.1070(+0.1219) | 0.2652(+0.2737) |
| Human(expert) | - | - | 62.67% | - | - | - |

## 4.2 Experimental results

As shown in Table 1, we compare the performance over a series of MLLMs by metrics including accuracy, IoU [18], GIoU [56] and DIoU [57]. The results indicate a general performance bottleneck for MLLMs on IAD tasks, characterized by accuracy rates typically ranging from 20% to 60% and significantly low IoU scores. Specifically, the commercial MLLMs GPT-4o and Gemini-2.0-flash achieve accuracy rates of only 49.19% and 59.64% respectively, while the average accuracy of pre-trained open-source models is even lower, at only 45.55%. Critically, the IoU scores for these MLLMs are near zero. These deficiencies suggest that MLLMs, when relying solely on their general visual capabilities, exhibit limitations in accurately identifying the defect types and the precise coordinates of anomalous regions.

However, there are significant performance improvement when trained on AnomalyCoT. Following LoRA fine-tuning, Llama 3.2-Vision achieves the highest accuracy of 94.02%, while Qwen2.5-VL attains the highest IoU score of 0.3225. Compared to baseline models, the average accuracy rate and IoU score increase by 36.49% and 0.20 respectively. Furthermore, full fine-tuning results in even greater enhancements, with the highest accuracy reaching 97.46% and the highest IoU score reaching 0.4252. These results suggest that MLLMs can effectively learn the diverse scenarios represented in AnomalyCoT, leading to great performance of anomaly detection and location. Moreover, compared with LoRA, the further performance improvement observed through full fine-tuning indicates that the diversity of our dataset scenarios do not lead to overfitting issues. The comprehension ability of the visual model is the basis of multimodal tasks, and the addition of reasoning ability of the language model has made a great contribution to the outstanding performance in IAD tasks. To contextualize the achieved accuracy, we further conducted a human evaluation by inviting an industrial anomaly detection researcher to answer 7,916 test questions, choosing the corresponding options according to our pictures and questions, just like MLLM, and then calculate the accuracy. This process took us 4 hours. As shown in Table 1, the human expert achieved 62.67% accuracy, outperforming pre-trained models but still lower than the best fine-tuned model (97.46%). This result highlights while human experts remain strong at zero-shot reasoning, the fine-tuned MLLMs surpass human-level performance after learning CoT reasoning, demonstrating the practical value of our dataset.

To further evaluate the generalization ability of our dataset, we conducted a cross-dataset validation where specific datasets were held out during training. Specifically, we excluded MVTec AD, MVTec

Table 2: Ablation experiment results.

| Model | Scale | Accuracy w.o. coordinate | Accuracy w.o. cot | IoU w.o. cot | GIoU w.o. cot | DIoU w.o. cot |
|---|---|---|---|---|---|---|
| InternVL 3 | 8B | 90.42%(+33.09%) | 62.25%(+4.92%) | 0.2653(+0.2248) | 0.0329(+0.1488) | 0.1778(+0.2050) |
| LLaVA-1.5 | 7B | 59.35%(+30.51%) | 64.45%(+35.61%) | 0.2631(+0.2631) | 0.027(+0.0270) | 0.1792(+0.1792) |
| Llama 3.2-Vision | 11B | **93.24%**(+37.17%) | **86.27%**(+30.20%) | 0.2528(+0.2527) | 0.0833(+0.0943) | 0.1982(+0.2036) |
| Qwen2-VL | 7B | 86.66%(+35.05%) | 80.99%(+29.38%) | **0.3652**(+0.3384) | **0.1779**(+0.2188) | **0.3102**(+0.3125) |
| Qwen2.5-VL | 7B | 89.70%(+44.17%) | 82.71%(+37.18%) | 0.3164(+0.2225) | 0.1701(+0.0950) | 0.2929(+0.2065) |
| Gemma 3 | 4B | 64.42%(+42.27%) | 53.15%(+31.00%) | 0.1147(+0.1144) | -0.1228(-0.0990) | 0.0417(+0.0558) |
| Gemma 3 | 12B | 53.51%(+14.39%) | 62.12%(+22.00%) | 0.2088(+0.2086) | -0.0178(-0.0029) | 0.1274(+0.1359) |

**Notations:** *w.o.* stands for without.

LOCO AD, VisA, and GoodsAD respectively, and the model was fine-tuned exclusively on the remaining datasets and tested during inference on these previously unseen scenarios. The results demonstrate that the model fine-tuned with CoT annotations consistently maintained strong performance on unseen datasets, significantly outperforming fine-tuning without CoT. LLMs unlearning method[58] give us inspiration. Detailed results are reported in Appendix H.

## 4.3 Ablation Study

The responses in AnomalyCoT consist of CoT reasoning and the coordinates of defects. To comprehensively understand the contributions to the great performance of MLLMs, we conduct ablation experiments for these two factors. Based on AnomalyCoT, we construct two ablation datasets. One has no coordinate in analysis process and the other has no specific reasoning but only answers and coordinates. We conduct LoRA fine-tuning for open-sources MLLMs in these two datasets and the results are recorded in Table 2.

**Fine-tuning without CoT.** In this study, we investigate the effect of natural language reasoning in combination with visual features in anomaly detection. When removing specific reasoning process, MLLMs can only learn from the direct mapping from the original image and question to the correct option and coordinates. The performances of MLLMs are all greatly improved compared to baselines, but they are still much lower than the results of models fine-tuned on AnomalyCoT. This suggusts that through the correct option and the coordinates of anomaly regions, MLLMs can learn effective knowledge of anomaly detection tasks while the CoT reasoning can provide further comprehension of the question and image.

**Fine-tuning without coordinate.** In the ablation of coordinates, we reconstruct the related sentences or provide general orientation to ensure there is only a difference in location information compared with AnomalyCoT. This dataset can still guide MLLMs to reason based on the question and the image. The performance of each model have declined on the whole compared to AnomalyCoT, but the results vary according to the different models. For MLLMs including LLaVA-1.5-7B and Gemma3-4B, the performances of the ablation experiment are a little higher than AnomalyCoT. This indicates that different models have different sensitivity to textual interpretation and numerical coordinates and learning and fitting capabilities. We innovatively introduced CoT reasoning in the multimodal anomaly detection task and provided the accurate coordinates of the anomaly region in CoT, which is of great benefit to the performance of the anomaly detection task.

## 4.4 Multi-scenario Analysis

The unbalanced number of samples has always been an important problem in the training of deep learning models [59], and it may lead to overfitting and low confidence. Since AnomalyCoT consists of images with different sample numbers from different industrial scenarios, we verify whether it has this problem. Table 3 shows the performance of Llama 3.2-Vision in 5 main scenarios, and the results of all scenarios are recorded in Appendix G.

Firstly, the trend in fine-tuning efficacy across individual scenarios aligns consistently with the overall results, with full fine-tuning outperforming LoRA fine-tuning and in turn outperforming the pre-trained model performance. This consistency demonstrates that the results of our experiment

Table 3: Results of Llama 3.2-Vision in 5 main scenarios.

| Scenario | Type | Accuracy | IoU | GIoU | DIoU |
|---|---|---|---|---|---|
| GoodsAD 2962 | Pre-trained | 54.21% | 0.0000 | 0.0000 | 0.0000 |
| | LoRA | 90.07%(+35.86%) | 0.0184(+0.0184) | 0.0174(+0.0174) | 0.0178(+0.0178) |
| | Full | 97.31%(+43.10%) | 0.0339(+0.0339) | 0.0318(+0.0318) | 0.0321(+0.0321) |
| MIAD 16725 | Pre-trained | 49.03% | 0.0000 | -0.8973 | -0.4190 |
| | LoRA | 96.23%(+47.20%) | 0.3473(+0.3473) | 0.0030(+0.9003) | 0.2190(+0.6380) |
| | Full | **97.73%**(+48.70%) | **0.3907**(+0.3907) | 0.0682(+0.9655) | **0.2834**(+0.7024) |
| MVTec-LOCO 1427 | Pre-trained | 73.08% | 0.0000 | 0.0000 | 0.0000 |
| | LoRA | 92.66%(+19.58%) | 0.0393(+0.0393) | -0.0031(-0.0031) | 0.0046(+0.0046) |
| | Full | 94.06%(+20.98%) | 0.0497(+0.0497) | 0.0171(+0.0171) | 0.0235(+0.0235) |
| MVTecAD 2906 | Pre-trained | 68.95% | 0.0000 | -0.2702 | -0.1222 |
| | LoRA | 87.65%(+18.70%) | 0.2068(+0.2068) | 0.1455(+0.4157) | 0.1717(+0.2939) |
| | Full | 97.26%(+28.31%) | 0.2223(+0.2223) | **0.1661**(+0.4363) | 0.1931(+0.3153) |
| VisA 2381 | Pre-trained | 64.50% | 0.0000 | 0.0000 | 0.0000 |
| | LoRA | 89.50%(+25.00%) | 0.0110(+0.0110) | -0.0353(-0.0353) | -0.0230(-0.0230) |
| | Full | 94.33%(+29.83%) | 0.0216(+0.0216) | -0.0175(-0.0175) | -0.0033(-0.0033) |

are universal in each scenario, and resolves concerns about potential biases caused by the imbalance of samples in the sub-datasets. Notably, the accuracy rates of the pre-trained model on MIAD and MVTec-LOCO are 49.03% and 73.08% respectively, with a gap of 24.05%. However, after fine-tuning, the accuracy rates on both sets reach over 90%, with a gap of less than 4%. Besides, Figure 14 visualizes the responses of Llama 3.2-Vision in two scenarios, showing that the model can extract different features for different objects and conduct different analyses.

Furthermore, from the result data of the ablation experiments in various scenarios, we found that after removing the thought chain, the accuracy of some scenarios decreased significantly (such as VisA), and after removing the regional coordinates, the accuracy of some scenarios decreased significantly (such as Nanotwice), and most scenarios belonged to the former. This indicates that in multiple scenarios of actual industrial anomaly detection, it is necessary to provide both language and text explanations as well as the coordinates of the anomaly regions simultaneously in the training corpus.

## 5 Conclusion

In this paper, we briefly analyze the various limitations of previous anomaly detection datasets, particularly the lack of rigorous reasoning processes with precise visual information. To address these issues, we introduce AnomalyCoT, the first multimodal CoT dataset for the IAD task, which contains diverse scenarios and challenging tasks, providing strong support for the application of MLLMs in real-world environments to perform IAD tasks. In addition, we conduct different types of fine-tuning training on representative MLLMs, leveraging the CoT data to effectively learn professional thinking patterns across different scenarios. The experimental results demonstrate the reliability of the dataset, and all trained models achieve significant improvements in key metrics. Our results further confirm that incorporating CoT reasoning not only enhances interpretability but also substantially improves the generalization capability of fine-tuned models in cross-domain anomaly detection scenarios. Our contributions significantly advance the application of MLLMs in real-scene anomaly detection, establishing a solid foundation for future related studies.

One limitation of our dataset is that the initial CoT annotations were primarily generated using QwenVL-Max, which may introduce a subtle bias toward the reasoning style of this model family. Although manual verification and the strong performance of fine-tuned Llama-based models mitigate this concern, a broader exploration of diverse CoT generation sources would further improve robustness. We leave this as an interesting direction for future work.

## Acknowledgements

This work was supported by the National Key R&D Program of China (Grant No. 2022ZD0162000), the National Natural Science Foundation of China (Grant No. 62222211), and the NSFC Joint Fund for Research on Lightweight and Autonomous Intelligence for Cloud-Edge Collaboration in Content Generation (Grant No. U24A20326).

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

# A    AnomalyImage

Table 4 presents the information of AnomalyCoT in more detail, including the objects contained in each sub-dataset and the corresponding defect types. It also records the specific number of samples corresponding to different objects and different numbers of defects.

Table 4: Construction of AnomalyImage.

| Dataset | Object | Number | Sample | | | Defect types |
|---|---|---|---|---|---|---|
| | | | single | multiple | no | |
| MVTecAD 2906 | bottle | 174 | 63 | - | 111 | broken, contamination |
| | cable | 231 | 80 | 10 | 141 | bent wire, cable swap, cut, missing, poke insulation |
| | capsule | 229 | 109 | - | 120 | crack, faulty imprint, poke, scratch, squeeze |
| | carpet | 231 | 80 | - | 151 | color, cut, metal contamination, thread |
| | hazelnut | 212 | 58 | - | 154 | crack, cut, hole, print |
| | leather | 226 | 85 | - | 141 | color, cut, fold, glue, poke |
| | metal nut | 211 | 90 | - | 121 | bent, color, flip, scratch |
| | pill | 284 | 123 | 16 | 145 | color, contamination, crack, faulty imprint, type, scratch |
| | screw | 296 | 115 | - | 181 | manipulated front, scratch, thread |
| | tile | 213 | 83 | - | 130 | crack, glue strip, gray stroke, oil, rough |
| | transistor | 179 | 38 | - | 141 | bent lead, cut lead, damaged case, misplaced |
| | wood | 185 | 46 | 9 | 130 | color, hole, liquid, scratch |
| | zipper | 235 | 102 | 13 | 120 | broken, fabric, rough, split, squeezed |
| MVTec-LOCO 1427 | breakfast box | 260 | 158 | - | 102 | logic, structure |
| | juice bottle | 283 | 189 | - | 94 | logic, structure |
| | pushpins | 308 | 170 | - | 138 | logic, structure |
| | screw bag | 340 | 218 | - | 122 | logic, structure |
| | splicing connectors | 236 | 117 | - | 119 | logic, structure |
| MSD | phone | 1220 | 1200 | - | 20 | oil, scratch, stain |
| MPDD 434 | bracket black | 73 | 47 | - | 26 | hole, scratch |
| | bracket brown | 82 | 50 | - | 32 | parts mismatch |
| | bracket white | 60 | 30 | - | 30 | defective printing, scratch |
| | connector | 22 | 13 | - | 9 | parts mismatch |
| | metal plate | 96 | 70 | - | 26 | rust, scratch |
| | tubes | 101 | 69 | - | 32 | color, crush-bend cutting, cut, deformation, flat crush, |
| MIAD 16725 | catenary dropper | 3054 | 2426 | - | 628 | broken, looseness, miss |
| | electrical insulator | 2868 | 2025 | - | 843 | broken |
| | nut and bolt d | 2272 | 1592 | - | 680 | missnut, looseness |
| | photovoltaic module | 3473 | 2051 | - | 1422 | broken, foreign body, miss |
| | wind turbine | 1994 | 1501 | - | 493 | crack |
| | witness mark | 3064 | 1993 | - | 1071 | looseness |
| GoodsAD 2962 | cigarette box | 426 | 243 | - | 183 | opened |
| | drink bottle | 772 | 417 | - | 355 | cup half open, cup open, surface damage |
| | drink can | 292 | 146 | - | 146 | deformation, strew missing, surface damage |
| | food bottle | 599 | 356 | - | 243 | deformation, opened, surface damage |
| | food box | 391 | 247 | - | 144 | deformation, opened, surface damage |
| | food package | 482 | 229 | - | 253 | broken, surface anomaly |
| Defect Spectrum 1197 | DS-Cotton-Fabric | 102 | 86 | - | 16 | bubble, texture |
| | DS-DAGM | 330 | 266 | - | 64 | color, crush, dirty, scratch, texture |
| | DS-VISION | 765 | 589 | 20 | 156 | scratch, crack, crush, bump, dirty, gap, friction, texture, color, fiber, point |
| VisA 2381 | candle | 197 | 94 | 3 | 100 | bump, dent, deposit, scratch, uneven edge, wick |
| | capsules | 199 | 99 | - | 100 | blister, flattening, inclusion, leakage, scratch |
| | cashew | 187 | 92 | - | 95 | adhesion, coating residue, discoloration, edge chip, hole, inclusion, pinhole, scratch, surface damage |
| | chewinggum | 199 | 99 | - | 100 | contamination, corner chip, scratch, surface damage |
| | fryum | 200 | 80 | 20 | 100 | break, deposit, scratch |
| | macaroni1 | 200 | 100 | - | 100 | chip, crack, discoloration, hole, scratch, stain |
| | macaroni2 | 200 | 100 | - | 100 | chip, crack, discoloration, hole, scratch, stain |
| | pcb1 | 200 | 100 | - | 100 | lead bend, miss, scratch, solder residue |
| | pcb2 | 205 | 100 | - | 105 | lead bend, miss, scratch, solder residue |
| | pcb3 | 198 | 98 | - | 100 | lead bend, miss, scratch, solder residue |
| | pcb4 | 197 | 97 | - | 100 | component damage, foreign material, miss, oxidation, scratch |
| | pipe_fryum | 199 | 99 | - | 100 | break, deposit, overlap, scratchm, stain |
| NanoTwice | NanoTwice | 49 | 35 | - | 14 | dust |
| Self 8466 | Car | 2979 | 2008 | 8 | 963 | break, crack, dent, surface damage |
| | Road | 2247 | 2162 | - | 85 | crack |
| | Wood | 194 | 194 | - | - | crack, scratch |
| | Fabric | 3046 | 2272 | 280 | 494 | color deviation, crease, hole, seam mark, stain, watermark, weaving defect |

# B  Prompt

---

**Prompt for CoT generation**

# role definition
You are an AI model for anomaly detection. You should provide a reasoning process for the following question as instructed.
# correct answer
The correct answer is \<option\> and the anomaly region coordinates \<boxes\>.
# reasoning
Please analyze the entire image to support the correct choice. For each incorrect option, provide a rejection reason citing specific missing features. If the answer is \"no defect\", please provide a brief analysis; If not,immediately specify the anomaly region coordinates and analyse this region in detail. Then state \"The defect type is  \<type\>\", appending \"Likely cause:\" with a explanation. Finally, conclude your response with \"The answer is \<option\>.\". The coordinates are given in pixels, the anomaly region is defined by its top-left and bottom-right corners.

---

Figure 4: Prompt for CoT generation. In the generation of CoT, we provide Qwen-VL-MAX with the correct choice and the anomaly region coordinates if existing. For different questions, these information will be replaced with the corresponding ones, that is, <\option> and <\boxes> in the figure.

---

**Prompt for training and evaluation**

# role definition
You are an AI model for anomaly detection. You should answer the following question as instructed.
# reasoning
Please analyze the entire image to support your judgment. For each incorrect option, provide a rejection reason citing specific missing features. If your answer is \"no defect\", please provide a brief analysis; If not, immediately specify the anomaly region coordinates in the format of [[(x1, y1),(x2, y2)], ... ] and analyse this region in detail to identify the defect type. Finally, conclude your response with \"The answer is \<option\>.\".

---

Figure 5: Prompt for training and evaluation. This prompt is used in the fine-tuning on AnomalyCoT and the evaluation of MLLMs, and it guides MLLMs to reason out the defect types and coordinates.

---

**Prompt for ablation of coordinate**

# role definition
You are an AI model for anomaly detection. You should answer the following question as instructed.
# reasoning
Please analyze the entire image to support your judgment. For each incorrect option, provide a rejection reason citing specific missing features. If the answer is \"no defect\", please provide a brief analysis; If not, analyse the anomaly region in detail to identify the defect type. Finally, conclude your response with \"The answer is \<option\>.\".

---

Figure 6: Prompt for ablation of coordinate. This prompt is used to fine-tune MLLMs in the ablation study for coordinate.

---

**Prompt for ablation of CoT**

# role definition
You are an AI model for anomaly detection. You should answer the following question and identify the anomaly region coordinates if existing.

---

Figure 7: Prompt for ablation of CoT. This prompt is used to fine-tune MLLMs in the ablation study for reasoning process.

## C Data Structure

Each sample in AnomalyCoT is organized in the structure shown Figure 8. It contains two main part, messages and images. In messages there are system prompt, user query and model response. In images there are input images.

```
{
   "messages": [
     {
       "content": "You are an AI model for anomaly detection. You should......",
       "role": "system"
     },
     {
       "content": "<image>Which of the following options correctly identifies the defects in the image? ......",
       "role": "user"
     },
     {
       "content": "My analysis based on this image is as follows: ......",
       "role": "assistant"
     }
   ],
   "images": [
     "AnomalyCOT/MIAD/wind_turbine/crack/0000.jpg"
   ]
}
```

Figure 8: Data structure of AnomalyCoT.

## D Fine-Tuning Configuration

We conducted LoRA and full-parameter fine-tuning on diverse MLLMs using 8 NVIDIA A100 GPUs, and the hyperparameter settings used during the fine-tuning process are detailed in Table 5.

Table 5: Hyperparameter Settings for Fine-tuning.

| Hyperparameter | Value / Strategy |
| --- | --- |
| Batch size | 32 |
| Cutoff length | 8192 |
| Optimizer | AdamW |
| Initial learning rate | 1e-4 |
| Learning rate scheduler | Cosine decay |
| Warmup ratio | 0.1 |
| Precision | BF16 |
| Number of epochs | 3 |

# E Sample of logical error

The filtering of CoT consists of two steps. The first is the filtering of the format and the second is the filtering of the logic. Logic error includes various forms such as causal transposition and semantic contradictions, Fig 9 is a specific sample of causal transposition.

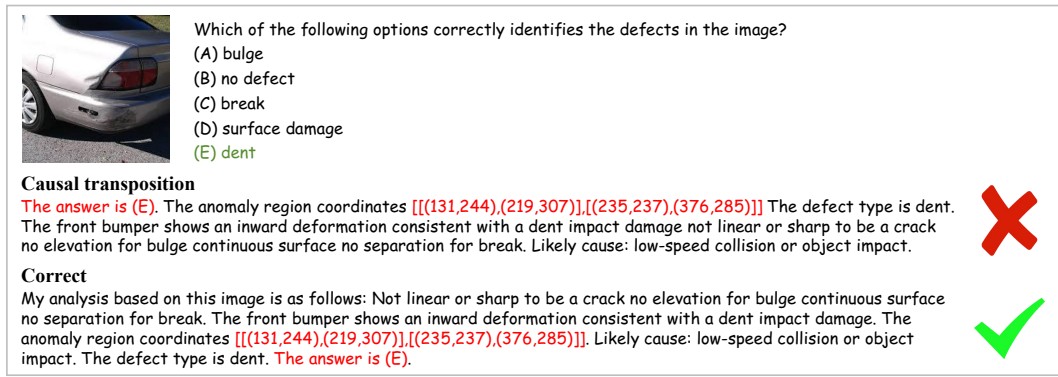

Figure 9: Sample of causal transposition. The reversal and effect is a typical error of the generated CoT data. To guarantee the quality of AnomalyCoT, we filter and recycle these erroneous data. This figure shows the original CoT with logic error and the correct CoT after recycling.

# F Case Study

We demonstrate the responses of different models in this section. Figure 10,11,12 report the responses of different fine-tuned models based on Qwen2.5-VL-7B of questions of different difficulties. Figure 13 reports the responses of different MLLMs after LoRA fine-tuning on AnomalyCoT.

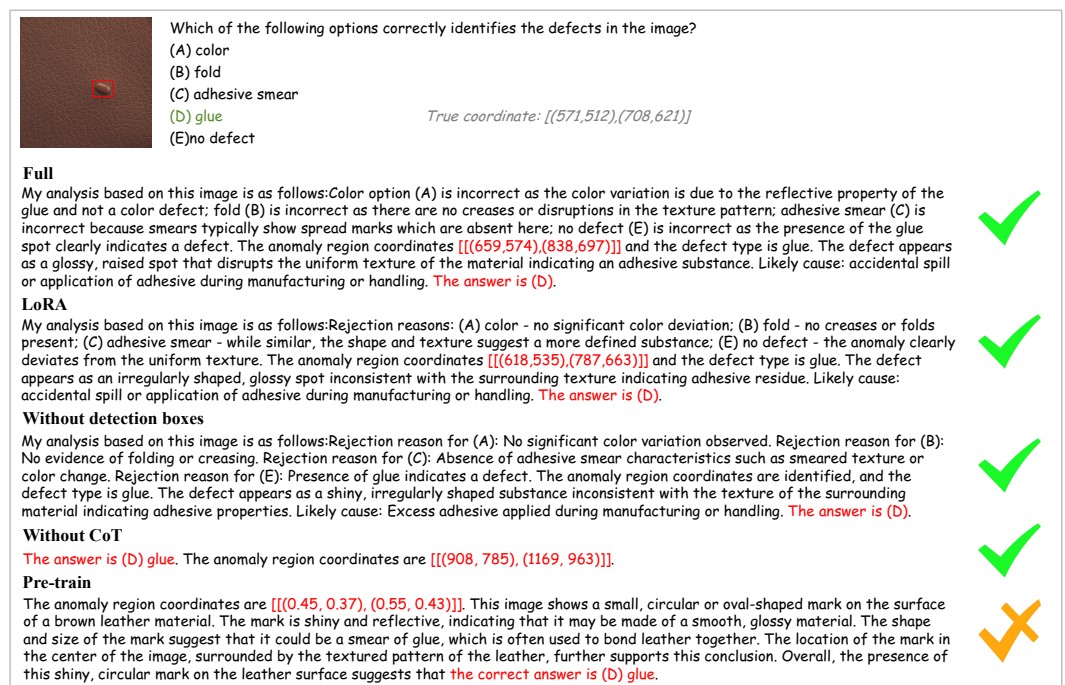

Figure 10: Comparison between different fine-tuned models based on Qwen2.5-VL-7B. The photo shows a leather surface with glue. All the fine-tuned models can select the correct defect type and identify the coordinates of the defect.

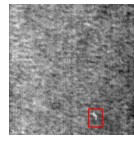

Which of the following options correctly identifies the defects in the image?
(A) no defect
(B) bump
(C) crush
(D) shade
(E) color                    *True coordinate: [(310,417),(351,458)]*

**Full**
My analysis based on this image is as follows:For option (A), no defect is incorrect as there is a visible anomaly. Option (B), bump, is rejected because there is no evidence of a raised surface; the texture remains consistent except for the color variation. Option (C), crush, is incorrect as there are no signs of compression or distortion in the texture. Option (D), shade, is rejected because the anomaly is not a shadow but a change in color. The defect type is color. The anomaly region coordinates [[(317,415),(349,444)]] show a noticeable variation in color compared to the surrounding area indicating a color defect. Likely cause: The color variation could be due to a material impurity or a manufacturing defect affecting the dyeing process. The answer is (E). 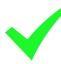

**LoRA**
My analysis based on this image is as follows:Option (A) no defect rejected because there is visible irregularity. Option (B) bump rejected because there is no raised area. Option (C) crush rejected because there is no indentation or compression. Option (D) shade rejected because lighting variations are consistent across the image. The anomaly region coordinates [[(329,416),(378,459)]] show a lighter spot compared to the surrounding area indicating an inconsistency in color. Likely cause: uneven dye application or material inconsistency. The answer is (E). The defect type is color. 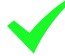

**Without detection boxes**
My analysis based on this image is as follows:Option (A) rejected as there are visible defects. Option (B) rejected as there's no raised surface indicative of a bump. Option (C) rejected as there's no compression or deformation suggesting a crush. Option (E) rejected as the color variation is consistent with the texture and doesn't indicate a defect. The anomaly region coordinates are identified in the image. The defect type is shade. The specified area shows a darker patch compared to the surrounding uniform texture indicating a shading issue. Likely cause: uneven lighting or shadowing during capture. The answer is (D). 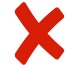

**Without CoT**
The answer is (C)crush. The anomaly region coordinates are [[(345, 416), (378, 452)]]. 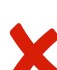

**Pre-train**
Analysis: The image appears to be uniformly gray without any visible defects such as bumps, crushes, shadows, or color variations. It seems to be a consistent background pattern with no anomalies detected. The answer is (A) no defect. 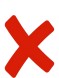

Figure 11: Comparison between different fine-tuned models based on Qwen2.5-VL-7B. The photo shows a textile surface with color inconsistence. Only the models fine-tuned on complete CoT data can select the correct defect type and identify the coordinates of the defect.

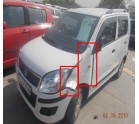

Which of the following options correctly identifies the defects in the image?
(A) break
(B) dent        *True coordinate: [[(347,178),(284,301)],[(284,257),(233,369)]]*
(C) bulge
(D) no defect
(E) surface damage

**Full**
My analysis based on this image is as follows:Rejection reasons: (A) break - no visible cracks or separation of material; (C) bulge - the deformation is inward not outward; (D) no defect - the dent is clearly visible; (E) surface damage - while present, it's secondary to the dent and not the primary defect. Anomaly region coordinates [[(194,295),(268,372)],[(276,216),(345,303)]] indicate a noticeable inward deformation on the car's front left side consistent with a dent. Likely cause: impact from another object causing the metal to deform inward without breaking. The answer is (B). The defect type is dent. 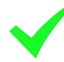

**LoRA**
My analysis based on this image is as follows:For option (A), there's no visible break as the surface continuity is intact without any cracks or separations. Option (C) is incorrect because a bulge would show an outward deformation which isn't observed here. Option (D) is rejected as the presence of a dent indicates a defect. Option (E) is not accurate since surface damage typically refers to scratches or paint issues rather than structural deformations. The anomaly region coordinates [[(345,165),(257,303)]] and the defect type is dent. The front left side of the car shows a noticeable inward deformation consistent with a dent which disrupts the smooth contour of the vehicle's bodywork. Likely cause: impact from another object causing the metal to deform inwardly. The answer is (B). 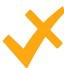

**Without detection boxes**
My analysis based on this image is as follows:Rejection reasons: (A) break - no visible cracks or separation; (C) bulge - no outward protrusion; (D) no defect - clear presence of deformation; (E) surface damage - no scratches or paint disruption. The anomaly region coordinates correspond to the identified area and the defect type is dent. The front left side of the car shows an inward deformation consistent with a dent likely caused by a low-speed impact or object strike. Likely cause: accidental collision or contact with another object. The answer is (B). 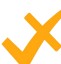

**Without CoT**
The answer is (B)dent. The anomaly region coordinates are [[(270, 235), (318, 348)]]. 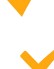

**Pre-train**
The answer is (D) no defect. There are no visible signs of break, dent, bulge, or surface damage in the image of the vehicle. The vehicle appears to be in good condition without any noticeable anomalies. 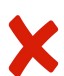

Figure 12: Comparison between different fine-tuned models based on Qwen2.5-VL-7B. The photo shows a car with dents. All the fine-tuned models can select the correct defect type, but only full-parameters fine-tuned model identify the coordinates of the defect in an expected degree of detail.

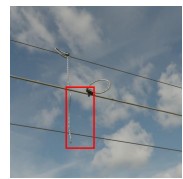

Which of the following options correctly identifies the defects in the image?
(A) paint defect
(B) no defect
(C) broken          *True coordinate: [[(128,119),(201,416)]]*
(D) deformation
(E) miss

**InternVL3-8B**

My analysis based on this image is as follows:Rejection reasons: (A) paint defect - no paint present on the wire; (B) no defect - the break is evident; (D) deformation - the wire is not bent or deformed but broken; (E) miss - the defect is clearly visible. The anomaly region coordinates [[(129,114),(198,419)]] and the defect type is broken. The wire shows a clear break with separated ends indicating discontinuity in the cable structure. Likely cause: mechanical stress or material fatigue leading to failure. The answer is (C).

**LLaVA-1.5-7B**

My analysis based on this image is as follows:Rejection reasons: (A) paint defect - no visible paint on wires; (B) no defect - clear visual evidence of breakage; (D) deformation - the wire is not bent or deformed but severed; (E) miss - the defect is present and identifiable. The anomaly region coordinates [[(121,105),(216,424)]] and the defect type is broken. The wire shows a clear break with separated ends indicating discontinuity in the line. Likely cause: mechanical stress or damage leading to snapping of the wire. The answer is (C).

**Llama 3.2 Vision-11B**

My analysis based on this image is as follows:Option (A) paint defect rejected as there are no signs of paint on the wire. Option (B) no defect rejected because the break is evident. Option (D) deformation rejected as the wire shows a clean break rather than a bent or twisted shape. Option (E) miss rejected since the defect is present and identifiable. The anomaly region coordinates [[(119,123),(210,453)]] and the defect type is broken. The wire shows a clear break with separated ends indicating discontinuity in the line. Likely cause: mechanical stress or damage leading to snapping of the wire. The answer is (C).

**Qwen2-VL-7B**

My analysis based on this image is as follows:Rejection reasons: (A) paint defect - no visible paint on the wire; (B) no defect - the break is evident; (D) deformation - the wire is not bent or deformed but broken; (E) miss - the defect is present and identifiable. The anomaly region coordinates [[(131,113),(191,416)]] and the defect type is broken. The wire shows a clear break with separated ends indicating discontinuity in the structure. Likely cause: mechanical stress or material fatigue leading to failure. The answer is (C).

**Qwen2.5-VL-7B**

My analysis based on this image is as follows:Rejection reasons: (A) paint defect - no paint visible on wires; (B) no defect - the break is evident; (D) deformation - the wire is not bent or twisted but severed; (E) miss - there is a clear defect present. The anomaly region coordinates [[(132,108),(196,419)]] and the defect type is broken. The wire shows a clear break with separated ends indicating discontinuity in the line. Likely cause: mechanical stress or material fatigue leading to failure at this point. The answer is (C).

**Gemma 3-12B**

The wire shows a clear break with separated ends indicating discontinuity in the line. Likely cause: mechanical stress or material fatigue leading to fracture. The answer is (C)broken. Anomaly region coordinates [(175,200),(38,465)].

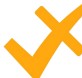

**Gemma 3-12B**

(A) paint defect - no visible paint on the cable; (B) no defect - the break is evident; (D) deformation - the shape of the cable is not altered except at the break; (E) miss - the defect is present and identifiable.  The anomaly region coordinates [[(113,124),(193,424)]] show a clear break in the cable with frayed edges indicating separation. Likely cause: mechanical stress or material fatigue leading to failure. The answer is (C).The defect type is broken.

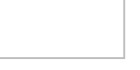

Figure 13: Comparison between different MLLMs. The photo shows a broken wire. Almost all the fine-tuned MLLMs can select the correct defect type and identify the coordinates of the defect.

# G  Muti-scenario Results

Table 6 shows the complete performance of Llama 3.2-Vision, containing all the fine-tuning methods, in all the scenarios (or sub-datasets). After fine-tuned on AnomalyCoT, the MLLM achieves great

performance on every scenario regardless of the number of corresponding samples in training set, which strongly proving the generalization ability of AnomalyCoT.

Table 6: Results of Llama 3.2-Vision in all scenarios

| Scene | Type | Accuracy | IoU | GIoU | DIoU |
|---|---|---|---|---|---|
| Defect Spectrum

samples: 237 | Pre-trained | 42.62% | 0.0000 | -0.8873 | -0.4237 |
| | LoRA | 85.65%(+43.03%) | 0.1166(+0.1166) | -0.0913(+0.7960) | 0.0383(+0.4620) |
| | Full | 92.83%(+50.21%) | 0.1851(+0.1851) | -0.0168(+0.8705) | 0.1270(+0.5507) |
| | w.o. CoT | 71.78%(+29.16%) | 0.1536(+0.1536) | -0.0544(+0.8329) | 0.0781(+0.5018) |
| | w.o. coordinate | 86.50%(+43.88%) | - | - | - |
| GoodsAD

samples: 594 | Pre-trained | 54.21% | 0.0000 | 0.0000 | 0.0000 |
| | LoRA | 90.07%(+35.86%) | 0.0184(+0.0184) | 0.0174(+0.0174) | 0.0178(+0.0178) |
| | Full | 97.31%(+43.10%) | 0.0339(+0.0339) | 0.0318(+0.0318) | 0.0321(+0.0321) |
| | w.o. CoT | 85.32%(+31.11%) | 0.0000(+0.0000) | 0.0000(+0.0000) | 0.0000(+0.0000) |
| | w.o. coordinate | 90.40%(+36.19%) | - | - | - |
| MIAD

samples: 3345 | Pre-trained | 49.03% | 0.0000 | -0.8973 | -0.4190 |
| | LoRA | 96.23%(+47.20%) | 0.3473(+0.3473) | 0.0030(+0.9003) | 0.2190(+0.6380) |
| | Full | 97.73%(+48.70%) | 0.3907(+0.3907) | 0.0682(+0.9655) | 0.2834(+0.7024) |
| | w.o. CoT | 89.57%(+40.54%) | **0.3997**(+0.3997) | 0.1083(+1.0056) | **0.3050**(+0.7240) |
| | w.o. coordinate | 95.87%(+46.84%) | - | - | - |
| MPDD

samples: 347 | Pre-trained | 59.65% | 0.0000 | -0.4992 | -0.3656 |
| | LoRA | 78.39%(+18.74%) | 0.1137(+0.1137) | -0.0201(+0.4791) | 0.0395(+0.4051) |
| | Full | 87.90%(+28.25%) | 0.1340(+0.1340) | 0.0208(+0.5200) | 0.0642(+0.4298) |
| | w.o. CoT | 70.89%(+11.24%) | 0.1064(+0.1064) | -0.0425(+0.4567) | 0.0296(+0.3952) |
| | w.o. coordinate | 75.22%(+15.57%) | - | - | - |
| MSD

samples: 244 | Pre-trained | 41.80% | 0.0000 | 0.0000 | 0.0000 |
| | LoRA | **100.00%**(+58.20%) | 0.0388(+0.0388) | -0.1101(-0.1101) | -0.0337(-0.0337) |
| | Full | 99.59%(+57.79%) | 0.0562(+0.0562) | -0.0707(-0.0707) | 0.0064(+0.0064) |
| | w.o. CoT | 99.18%(+57.38%) | 0.0817(+0.0817) | -0.0121(-0.0121) | 0.0518(+0.0518) |
| | w.o. coordinate | 98.77%(+56.97%) | - | - | - |
| MVTec-LOCO

samples: 286 | Pre-trained | 73.08% | 0.0000 | 0.0000 | 0.0000 |
| | LoRA | 92.66%(+19.58%) | 0.0393(+0.0393) | -0.0031(-0.0031) | 0.0046(+0.0046) |
| | Full | 94.06%(+20.98%) | 0.0497(+0.0497) | 0.0171(+0.0171) | 0.0235(+0.0235) |
| | w.o. CoT | 86.71%(+13.63%) | 0.0656(+0.0656) | 0.0398(+0.0398) | 0.0465(+0.0465) |
| | w.o. coordinate | 92.31%(+19.23%) | - | - | - |
| MVTecAD

samples: 583 | Pre-trained | 68.95% | 0.0000 | -0.2702 | -0.1222 |
| | LoRA | 87.65%(+18.70%) | 0.2068(+0.2068) | 0.1455(+0.4157) | 0.1717(+0.2939) |
| | Full | 97.26%(+28.31%) | 0.2223(+0.2223) | 0.1661(+0.4363) | 0.1931(+0.3153) |
| | w.o. CoT | 85.47%(+16.52%) | 0.3554(+0.3554) | **0.1683**(+0.4385) | 0.3040(+0.4262) |
| | w.o. coordinate | 89.54%(+20.59%) | - | - | - |
| NanoTwice

samples: 36 | Pre-trained | 33.33% | 0.0097 | -0.4297 | -0.1082 |
| | LoRA | 94.44%(+61.11%) | 0.0764(+0.0667) | -0.1915(+0.2382) | 0.0404(+0.1486) |
| | Full | 97.22%(+63.89%) | 0.2090(+0.1993) | -0.1016(+0.3281) | 0.1576(+0.2658) |
| | w.o. CoT | 100.00%(+66.67%) | 0.0952(+0.0855) | -0.0959(+0.3338) | 0.0597(+0.1679) |
| | w.o. coordinate | 83.33%(+50.00%) | - | - | - |
| VisA

samples: 476 | Pre-trained | 64.50% | 0.0000 | 0.0000 | 0.0000 |
| | LoRA | 89.50%(+25.00%) | 0.0110(+0.0110) | -0.0353(-0.0353) | -0.0230(-0.0230) |
| | Full | 94.33%(+29.83%) | 0.0216(+0.0216) | -0.0175(-0.0175) | -0.0033(-0.0033) |
| | w.o. CoT | 77.31%(+12.81%) | 0.0220(+0.0220) | 0.0086(+0.0086) | 0.0113(+0.0113) |
| | w.o. coordinate | 86.76%(+22.26%) | - | - | - |
| Self

samples: 1768 | Pre-trained | 59.79% | 0.0228 | -0.6799 | -0.3205 |
| | LoRA | 97.45%(+37.66%) | 0.2786(+0.2558) | 0.0658(+0.7457) | 0.2168(+0.5373) |
| | Full | 98.87%(+39.08%) | 0.2959(+0.2731) | 0.0642(+0.7441) | 0.2359(+0.5564) |
| | w.o. CoT | 85.47%(+25.68%) | 0.3554(+0.3326) | 0.1683(+0.8482) | 0.3040(+0.6245) |
| | w.o. coordinate | 96.21%(+36.42%) | - | - | - |

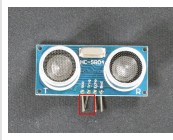
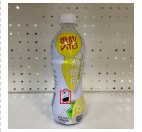

Which of the following options correctly
identifies the defects in the image?
(A) miss
(B) scratch
(C) dent
(D) lead bend
(E) no defect
*True coordinate: [(641,751),(690,888)]*

**Llama 3.2 Vision-11B**
My analysis based on this image is as follows:Rejection reason for (A):
no missing components or solder points visible. Rejection reason for
(B): no linear marks indicative of scratching. Rejection reason for (C):
no inward deformation suggesting a dent. Rejection reason for (E): the
bent lead is a clear defect. The defect type is lead bend at
coordinates [[(570,736),(700,895)]] as the lead appears bent and not
in its original straight position. Likely cause: mechanical stress or
improper handling during assembly or testing. The answer is (D).

Which of the following options correctly
identifies the defects in the image?
(A) surface damage
(B) no defect
(C) opened
(D) cap half open
(E) surface anomaly
*True coordinate: [(1160,1996),(1696,2428)]*

**Llama 3.2 Vision-11B**
My analysis based on this image is as follows:Rejection reasons: (B) no
defect - the image clearly shows an anomaly; (C) opened - the cap is
fully sealed; (D) cap half open - the cap is fully closed; (E) surface
anomaly - while there is a surface issue, it is specifically damage. The
defect type is surface damage. Anomaly region coordinates
[[(1485,1808),(1641,1932)]] indicate a visible scratch or abrasion on
the bottle's surface disrupting the label's integrity. This anomaly is
characterized by a disruption in the label's continuity and color
uniformity. Likely cause: mechanical impact or friction during handling
or packaging causing superficial damage to the label. The answer is (A).

Figure 14: Comparison of responses in different scenarios. MLLMs can make corresponding analyses for different image scenarios, and we present the responses from Llama 3.2-Vision (LoRA fine-tuned) in sub-datasets VisA and GoodsAD.

## H  Cross-validation

Table 7 presents the result of the result of our cross-validation process for fine-tuning.

Table 7: Result of cross-validation.

|  | CoT | w.o. CoT |
|---|---|---|
| Pre-trained | 46.18 | 35.75 |
| Pre-trained + dm | 38.12 | 29.32 |
| LoRA | 70.37 | 62.97 |
| LoRA+dm | 75.78 | 50.76 |

**Notations:** *dm* stands for domain knowledge.

## I  Comparison

Table 8 compares the results of our AnomolyCoT method with the traditional IAD method and other anomaly detection methods that support classification.

Table 8: Result of different anomoly detection methods.

| Method | MVTec-AD | |
|---|---|---|
|  | Image-AUC | Accuracy |
| SPADE | 85.42 | - |
| PaDiM | 90.56 | - |
| PatchCore | 98.81 | - |
| AnomalyDiffusion | - | 72.36 |
| DualAnoDiff | - | 78.81 |
| Llama 3.2-Vision (Pre-trained) | - | 68.95 |
| Llama 3.2-Vision (LoRA) | - | 87.65 |
| Llama 3.2-Vision (Full) | - | 97.26 |

