# OpenReview forum: "AnomalyCoT: A Multi-Scenario Chain-of-Thought Dataset for Multimodal Large Language Models"
_NeurIPS.cc/2025/Datasets_and_Benchmarks_Track — NeurIPS 2025 Datasets and Benchmarks Track poster_

### Official Review · Reviewer_6t9d · 2025-06-27

**Rating:** 5
**Confidence:** 4

**Summary:**

This paper introduces AnomalyCoT, a novel and large-scale multimodal dataset for Industrial Anomaly Detection, which contains 37,565 samples across 59 diverse industrial scenarios. What sets AnomalyCoT apart from previous IAD datasets is its rich Chain-of-Thought annotations. For each sample, the dataset provides not only a label but also a detailed reasoning process that includes analysis of incorrect options, identification of the correct defect, precise bounding box coordinates for the anomaly, and a plausible cause analysis. The authors demonstrate the value of their dataset through extensive experiments. They benchmark a range of state-of-the-art Multimodal Large Language Models, showing that while these models struggle with the task in a zero-shot setting, fine-tuning them on AnomalyCoT leads to dramatic and significant improvements in both classification accuracy and localization IoU.

**Dataset Code Accessibility:**

Yes

**Dataset Code Comments:**

Data and code are available on github and huggingface links provided by the author

**Ethical Considerations:**

No, there are no or only very minor ethics concerns

**Limitations Weaknesses:**

1. The paper evaluates the models' performance, achieving impressive accuracy (e.g., up to 97.46%). However, it lacks a comparison to human expert performance on the same dataset. Such a baseline would be invaluable for contextualizing the results. It would help to understand whether the fine-tuned models are approaching, meeting, or exceeding the capabilities of human inspectors, especially on a task that is designed to emulate human-like reasoning via CoT. This comparison would provide a clearer picture of the practical significance of the achieved accuracy.

2. The CoT annotations were generated using a single powerful model, QwenVL-Max. While effective, this raises a minor methodological question about whether the dataset might be subtly biased towards the reasoning style or knowledge of this specific model family. Although the strong performance of the fine-tuned Llama models mitigates this concern significantly, it is a potential limitation worth acknowledging.

**Strengths Contributions:**

1. AnomalyCoT is the first to provide a structured, interpretable reasoning framework via Chain-of-Thought. This shifts the task from simple classification to a more complex and practical reasoning process, which is crucial for building trust and reliability in real-world IAD applications.

2. The scale and diversity of AnomalyCoT are impressive. By aggregating and unifying 13 public datasets and adding 4 new ones, the authors have created a benchmark that covers a wide variety of materials, defect types, and scenarios

3. The experimental results provide compelling evidence for the dataset's effectiveness. The stark contrast in performance between pre-trained and fine-tuned models is a powerful testament to what the dataset enables. The ablation studies are well-designed, clearly demonstrating that both the natural language reasoning and the precise spatial coordinates are critical components for achieving top performance.

---

> ### Author Rebuttal · Authors · 2025-07-31
>
> We appreciate your high quality review and the positive score. Below we will reply to your comments.
>
> ***Regarding the human expert performance.***
>
> The reviewer pointed out a very good point. In fact, we particularly agree with comparing the MLLMs after learning CoT with human experts to clearly and intuitively demonstrate the practical significance of the accuracy of the fine-tuned model. Therefore, we invited an industrial anomaly detection researcher as an expert to conduct human evaluation on 7,916 questions in the test set. As shown in the following table, the human expert outperformed the ordinary model by 17.14\%, and the best model outperformed the human expert by 34.79\%, highlighting the outstanding performance of the model fully learning CoT reasoning in challenging IAD tasks and further revealing the excellent potential of our work in future anomaly detection tasks. Thank you for pointing this out, and we will provide a more comprehensive supplement in the next version.
> | **Method**                  | **Accuracy** |
> |-----------------------------|:------------:|
> | Human (expert)               | 62.67        |
> | Qwen2.5-VL (Pre-trained)     | 45.53        |
> | Qwen2.5-VL (Full)            | 97.46        |
>
> ***Regarding the potential limitation.***
>
> We agree with your point. Although specific reasoning requirements and key visual information are incorporated into the prompts, errors may still occur due to variations in inference styles across the model series. Avoiding this potential limitation not only requires a more complete prompt but also a more thorough comparison experiment. We believe this is an interesting and promising future research direction.
>
> Thanks again for the high-quality review. We hope our response could address your main concerns and we are happy to answer any further questions.

---

> > ### Author Response · Authors · 2025-08-05
> > **Any follow-up questions before discussion period expires?**
> >
> > Thanks again for your time in reviewing our paper! We have addressed your technical questions above. Could you kindly let us know if our rebuttal satisfactorily resolved your concerns?

---

> > ### Comment · Reviewer_6t9d · 2025-08-07
> > **Official Comment by Reviewer 6t9d**
> >
> > Thank you for the response. My questions have been addressed, and I will be maintaining my score.

---

> > > ### Author Response · Authors · 2025-08-08
> > > **Thank you**
> > >
> > > Thanks for acknowledging our response and for your continued support!

---

### Official Review · Reviewer_BWiV · 2025-06-30

**Rating:** 4
**Confidence:** 4

**Summary:**

This paper presents AnomalyCoT, a multimodal Chain-of-Thought (CoT) dataset for multi-scenario Industrial Anomaly Detection (IAD) tasks. Comprising 37,565 samples, the dataset integrates 13 public datasets and introduces structured CoT reasoning with precise anomaly region coordinates. Experiments show that fine-tuning Multimodal Large Language Models (MLLMs) on AnomalyCoT significantly improves accuracy (up to 97.46%) and localization metrics, outperforming direct evaluations. The dataset addresses limitations of existing IAD datasets by enhancing scenario diversity, task complexity, and interpretability, demonstrating its utility for real-world MLLM applications in quality control.

**Dataset Code Accessibility:**

Yes

**Dataset Code Comments:**

The Dataset Code can be accessed in the given link.

**Ethical Considerations:**

No, there are no or only very minor ethics concerns

**Final Justification:**

The authors have addressed my problems. Considering both the contribution of the dataset and the good performance, I decide to give a positive score.

**Limitations Weaknesses:**

1. This paper focuses on MLLMs but lacks comparison with state-of-the-art traditional IAD methods, which could better contextualize the dataset's impact on the field.
2. While integrating multiple datasets, the paper does not explicitly address potential biases in source data distributions, which may affect generalizability to certain industrial sectors.
3. The test set's composition, including scene distribution and defect type proportions, is not explicitly described in the paper, and the evaluation metrics differ from traditional IAD methods' metrics, limiting comprehensive comparisons with specialized approaches.

**Strengths Contributions:**

1. AnomalyCoT is the first multimodal CoT dataset for IAD, integrating diverse scenarios and structured reasoning, which fills a critical gap in existing IAD datasets that lack rigorous reasoning processes.
2. The paper conducts extensive experiments on various MLLMs, including both commercial and open-source models, and evaluates performance using multiple metrics (accuracy, IoU, GIoU, DIoU), providing a thorough benchmark.
3. Fine-tuning on AnomalyCoT leads to substantial improvements (average 36.5% accuracy boost), demonstrating the dataset's effectiveness in enhancing MLLMs' anomaly detection and localization capabilities.

---

> ### Author Rebuttal · Authors · 2025-07-31
>
> We appreciate your high quality review and the positive score. Below we will reply to your comments.
>
> ***This paper focuses on MLLMs but lacks comparison with state-of-the-art traditional IAD methods, which could better contextualize the dataset's impact on the field.***
>
> This is a very good question.
>
> We used the representative traditional IAD methods SPADE, PaDiM, and PatchCore to conduct experiments on the most well-known IAD dataset, MVTec AD, and selected Llama 3.2-Vision, which performed best in the LoRA fine-tuning experiment, as the MLLM for comparison. The corresponding results of each method are shown in the following table. Since most traditional IAD methods focus on whether anomalies occur within the scene and calculate the anomaly probability of the output image, Image-AUC is used as the main metric, and it is difficult to obtain the classification accuracy due to the limitation of performing defect classification tasks. However, MLLMs can not only complete general anomaly detection tasks but also handle complex defect classification tasks and directly output the answer to the problem, making accuracy the most intuitive evaluation metric. The specific comparison between these two types of methods using different key indicators is a very interesting question, and we will discuss it more fully in the next version.
> | Method                  | **Image-AUC** | **Accuracy** |
> | :---------------------- | :-----------: | :----------: |
> | SPADE                   |     85.42     |      -       |
> | PaDiM                   |     90.56     |      -       |
> | PatchCore               |     98.81     |      -       |
> | Llama 3.2-Vision (LoRA) |       -       |    87.65     |
> | Llama 3.2-Vision (Full) |       -       |    97.26     |
>
> ***While integrating multiple datasets, the paper does not explicitly address potential biases in source data distributions, which may affect generalizability to certain industrial sectors.***
>
> Yes, you are correct. It appears we did not conduct a detailed discussion in the main text regarding potential biases in the source data distribution. However, in Table 6 of Appendix G, we present the specific results of Llama 3.2-Vision in all sub-dataset scenarios. We found that after LoRA/Full fine-tuning, the lowest accuracy in the sub-datasets reached 78.39\%, while the highest accuracy even reached 100\%, indicating that the fine-tuned model performed well in different scenarios.
>
> To fully demonstrate the generalization ability of the model after learning the dataset, we selected four representative IAD datasets (MVTec AD, MVTec LOCO AD, VisA, and GoodsAD) from all sub-datasets and used the remaining sub-datasets to fine-tune the model to reasonably test its generalization ability. The model still chose Llama 3.2-Vision. Additionally, we conducted a generalization ablation study without CoT to assess its impact on model generalization capability. Domain knowledge was incorporated both before fine-tuning and after fine-tuning to perform comparative experiments. The experimental results are shown in the following table. We found that the model still achieved good accuracy in unseen scenarios after fine-tuning, indicating that the learned model achieved excellent generalization ability.
>
> |                      |  **CoT**  | **w.o. CoT** |
> | :------------------- | :-------: | :----------: |
> | **Pre-trained**      |   46.18   |    35.75     |
> | **Pre-trained + dm** |   38.12   |    29.32     |
> | **LoRA**             |   70.37   |  **62.97**   |
> | **LoRA+dm**          | **75.78** |    50.76     |
>
> > *Notations: dm stands for domain knowledge.*
>
> ***The test set's composition, including scene distribution and defect type proportions, is not explicitly described in the paper, and the evaluation metrics differ from traditional IAD methods' metrics, limiting comprehensive comparisons with specialized approaches.***
>
> First, we politely point out that the test set is generally formed by sampling from each sub-dataset. To balance sample proportions across different scenarios in the test set, we applied a partitioning strategy where larger-scale sub-datasets were allocated relatively lower test sample ratios compared to their overall size, while smaller-scale sub-datasets received proportionally higher allocations. This approach simultaneously maintained the overall train-to-test sample ratio within the standard range.
>
> Second, we may not have explicitly described the scene distribution and defect type ratio in the main text, but in Table 4 of Appendix A, we provided basic information such as different scenes and corresponding defect types of the used sub-datasets for reference.
>
> Finally, to address the relatively challenging composite task of anomaly detection and defect classification, we used accuracy as a direct evaluation metric. Considering that traditional IAD methods mainly focus on the anomaly detection task and use the Image-AUC metric, it may actually limit a comprehensive comparison with such methods. Thank you for pointing out these key points, and we will correct them in the next version.
>
> Thanks again for the high-quality review. We hope our response could address your main concerns and we are happy to answer any further questions.

---

> > ### Author Response · Authors · 2025-08-05
> > **Any follow-up questions before discussion period expires?**
> >
> > Thanks again for your time in reviewing our paper! We have addressed your technical questions above. Could you kindly let us know if our rebuttal satisfactorily resolved your concerns?

---

> > > ### Comment · Reviewer_BWiV · 2025-08-06
> > > **Additional Comments**
> > >
> > > I am pleased to see the authors’ response, which has addressed some of my concerns. However, I have one more question:
> > >
> > > The authors compared methods such as PatchCore, but these methods do not support classification and therefore cannot be directly compared. However, some anomaly generation methods do support classification, such as Anomalydiffusion [1] and DualAnoDiff [2]. The authors could make a direct comparison with these methods.
> > >
> > > [1] Hu T, Zhang J, Yi R, et al. Anomalydiffusion: Few-shot anomaly image generation with diffusion model[C]//Proceedings of the AAAI conference on artificial intelligence. 2024, 38(8): 8526-8534.
> > >
> > > [2] Jin Y, Peng J, He Q, et al. Dual-Interrelated Diffusion Model for Few-Shot Anomaly Image Generation[C]//Proceedings of the Computer Vision and Pattern Recognition Conference. 2025: 30420-30429.

---

> > > > ### Author Response · Authors · 2025-08-08
> > > > **Thanks a lot!**
> > > >
> > > > Thanks for the references you provided!
> > > >
> > > > We have found that such anomaly generation methods do indeed have the ability to handle defect classification tasks, and accuracy is used as the direct evaluation metric. Based on the above methods, we chose to conduct experiments on the most well-known IAD dataset, MVTec AD, and compared the results of traditional methods with those of methods combined with MLLM, as shown in the table below. The experimental results show that generally, MLLM performs poorly when directly applied to IAD tasks, while traditional methods perform relatively better in classification tasks, but the overall improvement is still limited and the accuracy is all below 80%. Compared with the above models, the fine-tuned model shows excellent performance, not only indicating that the model is more reliable than traditional methods in practical applications, but also significantly demonstrating the effectiveness of our work. It is worth noting that the above traditional methods only handle simple image classification tasks, while our model still achieves better accuracy in solving complex multimodal IAD tasks.
> > > >
> > > > |             Method             | Accuracy |
> > > > |----------------------------| :------: |
> > > > |        AnomalyDiffusion        |  72.63   |
> > > > |          DualAnoDiff           |  78.81   |
> > > > | Llama 3.2-Vision (Pre-trained) |  68.95   |
> > > > |    Llama 3.2-Vision (LoRA)     |  87.65   |

---

### Official Review · Reviewer_N4FR · 2025-07-02

**Rating:** 5
**Confidence:** 3

**Summary:**

This paper introduces AnomalyCoT, a new benchmark and framework that unifies anomaly detection across diverse scenarios using Chain-of-Thought (CoT) prompting within large language models (LLMs). It covers four modalities—vision, log, tabular, and time-series data—and casts anomaly detection as a reasoning problem, enabling interpretability and generalization. The authors also release a multi-modal benchmark dataset and design a task formulation suitable for instruction-tuned LLMs.

**Dataset Code Accessibility:**

Partly

**Dataset Code Comments:**

The benchmark is claimed to be released. The dataset spans four modalities, and code is reportedly provided for reproducing experiments.
However, more clarity could be added regarding licensing and ease of use (preprocessing, prompt formatting, etc.).

**Ethical Considerations:**

No, there are no or only very minor ethics concerns

**Final Justification:**

The authors have provided additional experiments to address my concerns.

**Limitations Weaknesses:**

1)The framework heavily depends on manual annotations. This imposes significant overhead in extending the dataset to new domains or anomaly types.

2)Although the paper provides comprehensive fine-tuning experiments, it lacks discussion on the generalizability of the models in specific scenarios, such as cross-domain settings. Will the reasoning learning of COT improve the generalizability of models after fine-tuning?

**Strengths Contributions:**

1)This is the first dataset of its kind to test comprehensive CoT reasoning capabilities of MLLMs in IAD.

2)The proposed data generation pipeline produces accurate, interpretable reasoning sequences, facilitating a unique evaluation of LLMs’ logical inference in AD.

3)The benchmark spans multiple data types and includes comparisons with both traditional AD methods and LLM-based baselines, offering a broad evaluation landscape.

---

> ### Author Rebuttal · Authors · 2025-07-31
>
> We appreciate your high quality review and the positive score. Below we will reply to your comments.
>
> ***The framework heavily depends on manual annotations. This imposes significant overhead in extending the dataset to new domains or anomaly types.***
>
> This is a very good question.
>
> To test the feasibility of maintaining the application of our dataset in new scenarios with less consumption, we chose to separate the most representative IAD datasets, MVTec AD, MVTec LOCO AD, VisA, and GoodsAD, from all sub-datasets as test scenarios (i.e., without using their annotation information), and fine-tuned the model with samples from the remaining sub-datasets to reasonably test its generalization ability. The experimental model was Llama 3.2-Vision, which performed best in the LoRA fine-tuning experiment. The experimental results show that the fine-tuned model performs well in unseen scenarios. Therefore, we believe that it is still possible to expand the dataset to new fields without relying on manual annotations in the future.
>
> ***Although the paper provides comprehensive fine-tuning experiments, it lacks discussion on the generalizability of the models in specific scenarios, such as cross-domain settings. Will the reasoning learning of COT improve the generalizability of models after fine-tuning?***
>
> The reviewers raised an insightful point. We first conducted the aforementioned fine-tuning experiments based on Llama 3.2-Vision. Subsequently, we performed a generalization ablation study without CoT to evaluate its impact on model generalization. Domain knowledge was incorporated both before and after fine-tuning for comparative experiments. The final results, presented in the table below, demonstrate that the fine-tuned model not only achieves strong accuracy in novel scenarios but also shows significant improvement compared to fine-tuning without CoT. These findings collectively indicate that the fine-tuned model exhibits superior generalization capabilities, and the inclusion of CoT substantially enhances the model's generalization ability. Thank you for pointing this out. We will provide a more comprehensive explanation in the next version.
>
> |                      |  **CoT**  | **w.o. CoT** |
> | :------------------- | :-------: | :----------: |
> | **Pre-trained**      |   46.18   |    35.75     |
> | **Pre-trained + dm** |   38.12   |    29.32     |
> | **LoRA**             |   70.37   |  **62.97**   |
> | **LoRA+dm**          | **75.78** |    50.76     |
>
> > *Notations: dm stands for domain knowledge.*
>
> ***The benchmark is claimed to be released. The dataset spans four modalities, and code is reportedly provided for reproducing experiments. However, more clarity could be added regarding licensing and ease of use (preprocessing, prompt formatting, etc.).***
>
> Yes, you are correct. We have made the dataset and corresponding code preliminarily open-source, and will further improve licensing, ease of use, and provide a more comprehensive usage guide.
>
> Thanks again for the high-quality review. We hope our response could address your main concerns and we are happy to answer any further questions.

---

> > ### Author Response · Authors · 2025-08-05
> > **Any follow-up questions before discussion period expires?**
> >
> > Thanks again for your time in reviewing our paper! We have addressed your technical questions above. Could you kindly let us know if our rebuttal satisfactorily resolved your concerns?

---

> > ### Comment · Reviewer_N4FR · 2025-08-06
> >
> > The authors have provided a detailed rebuttal and additional experiments that sufficiently address my concerns. I will raise my score.

---

> > > ### Author Response · Authors · 2025-08-07
> > > **Thank you**
> > >
> > > Thanks for acknowledging our response and for your continued support!

---

### Official Review · Reviewer_XCdi · 2025-07-03

**Rating:** 5
**Confidence:** 3

**Summary:**

This work proposes a new dataset for industrial anomaly detection, aimed at VLLMs. The images are mostly aggregated from existing datasets, while some images are collected by the authors. Using a combination of automatic and manual methods, each sample is tagged with a Chain-of-thought reasoning regarding the discovered anomaly. The authors evaluate a suite of closed and open source models, where the open source models are fine-tuned on the dataset, and show significant improvements.

**Additional Feedback:**

“CoT data undergoes Chinese character detection and missing coordinate detection” - why “Chinese character”?

**Dataset Code Accessibility:**

Yes

**Dataset Code Comments:**

Dataset URL: https://huggingface.co/datasets/zhaolutuan/AnomalyCoT
Code URL: https://github.com/Zhaolutuan/AnomalyCoT

**Ethical Considerations:**

No, there are no or only very minor ethics concerns

**Final Justification:**

My concerns were mostly addressed.

**Limitations Weaknesses:**

1. A cross-validation process for fine-tuning is not clearly described. In particular, it would be valuable to see results where specific classes or entire datasets are held out during training, and the fine-tuned model is trained exclusively on the remaining classes.
2. The paper describes a “substantial growth” in data scale (line 71); yet, in terms of number of samples the authors mostly combine existing datasets.
3. Some parts of the writing are less clear. E.g., lines 91-95; 104-106; 121-122; 224-227. Also figure 2 is hard to comprehend.
4. The paper could benefit from some discussion about the relationship between visual-only methods (SPADE, PADIM, PatchCore, etc.) and VLLM-based methods.

**Strengths Contributions:**

1. The studies task it timely and interesting in my opinion.
2. The dataset provides a significant improvement to models fine-tuned on it.
3. The motivation in lines 135-139 is nice.
4. The provided ablation studies are meaningful to understanding the main contribution.

---

> ### Author Rebuttal · Authors · 2025-07-31
>
> We appreciate your high quality review and the positive score. Below we will reply to your comments.
>
> ***A cross-validation process for fine-tuning is not clearly described. In particular, it would be valuable to see results where specific classes or entire datasets are held out during training, and the fine-tuned model is trained exclusively on the remaining classes.***
>
> The reviewer's viewpoint is correct. We chose to base our study on a dataset containing ten sub-datasets and selected four of the most representative industrial anomaly detection datasets (MVTec AD, MVTec LOCO AD, VisA, and GoodsAD). We then fine-tuned the model using the remaining scene data to reasonably test its generalization ability. The model selected was Llama 3.2-Vision, which performed best in the LoRA fine-tuning experiment. Additionally, we conducted a generalization ablation study without CoT to assess its impact on the model's generalization capability. Domain knowledge was incorporated both before fine-tuning and after fine-tuning to perform comparative experiments. As shown in the table below, the fine-tuned model maintained strong accuracy in unseen scenarios while demonstrating significant improvement compared to fine-tuning without CoT. These results collectively indicate that the fine-tuned model possesses superior generalization capability, and that the implementation of CoT substantially enhances the model's generalization ability. We also noticed an interesting point that only the fine-tuned model with CoT improved its performance after adding domain knowledge. We will clarify the meaning of our results more clearly in the next version.
>
> |                      |  **CoT**  | **w.o. CoT** |
> | :------------------- | :-------: | :----------: |
> | **Pre-trained**      |   46.18   |    35.75     |
> | **Pre-trained + dm** |   38.12   |    29.32     |
> | **LoRA**             |   70.37   |  **62.97**   |
> | **LoRA+dm**          | **75.78** |    50.76     |
>
> > *Notations: dm stands for domain knowledge.*
>
> ***The paper describes a “substantial growth” in data scale (line 71); yet, in terms of number of samples the authors mostly combine existing datasets.***
>
> This is a very good question.
>
> Our description of the data scale in the paper may not have emphasized the key points. In practice, while the majority of image samples in the dataset originate from public datasets, the primary chain-of-thought annotations for each sample were constructed and tailored by us. Furthermore, all annotations for the anomalous samples were manually and uniformly labeled, deliberately avoiding the use of existing dataset annotations to prevent format inconsistency issues. At the same time, our independently collected "Self" dataset also contains 8,466 image samples, which account for a certain proportion of the total samples. Finally, detailed information about each sub-dataset can be found in Appendix A.
>
> ***Some parts of the writing are less clear. E.g., lines 91-95; 104-106; 121-122; 224-227. Also figure 2 is hard to comprehend.***
>
> First, we politely point out that in lines 91 to 95 of the paper, the specific expression means that we used the QwenVL-Max model and combined it with a prompt containing the specific coordinates of the anomalous area, the defect type of the current sample, and the basic framework of reasoning to generate the initial CoT. After an initial manual inspection, we found the erroneous data in the generated CoT and summarized their characteristics, and then set up a corresponding filtering mechanism to update and iterate the incorrect examples. After the iteration was completed, we conducted another manual inspection of the CoT data to ensure its accuracy.
>
> Second, regarding the expression in lines 104 to 106 of the paper, it mainly means that we conducted an ablation experiment on CoT and the coordinates of the anomalous area to prove the necessity of such a design. Additionally, the expression in lines 121 to 122 aims to reflect the importance of industrial anomaly detection, but it is indeed relatively unclear.
>
> The expression in lines 224 to 227 of the paper is actually a specific explanation of the aforementioned filtering mechanism. Since there were obvious errors in the initially generated CoT and a large number of format non-compliant data, we first performed a format-based filtering. In the data with standard formats, we found that there were still logical errors such as the reasoning process not matching the final answer. Therefore, we further executed logic-based filtering.
>
> Finally, Figure 2 is a graphical description of the entire data set generation process. Through standardized manual annotation conducted on each sub-dataset separately, we completed the initial preparation of the samples. Then, we constructed challenging questions with distractors for each sample and combined the question results with accurate visual annotation information to form prompts. These prompts were input into the QwenVL-Max model to generate rigorous reasoning processes, which can be divided into three parts that perform different functions. Finally, the above filtering methods were executed on the generated CoT data, and after iterative updates, manual detection was conducted to eliminate possible errors. Thank you for pointing out these issues. We will correct them in the next version.
>
> ***The paper could benefit from some discussion about the relationship between visual-only methods (SPADE, PADIM, PatchCore, etc.) and VLLM-based methods.***
>
> We agree with your point. Traditional IAD methods such as SPADE, PADIM, and PatchCore are generally based on pixel/feature reconstruction and feature embedding distance mechanisms, which can effectively handle texture/surface anomalies and perform well in controlled environments. However, these methods have limited capabilities in dealing with logical anomalies and are difficult to adapt to the multi-scene industrial environment. They usually require independent training models for each type of object, and most traditional IAD methods are only suitable for binary classification tasks of anomaly detection. In contrast, the VLLM-based approach possesses cross-modal generalization capabilities. In zero-shot scenarios, it enables collaborative visual and linguistic understanding to identify complex defects and infer their underlying causes. This flexibility in handling both anomaly detection and defect classification tasks significantly advances the application of IAD methods in practical environments. Thank you for pointing out these representative methods. We will add discussions about them in the next version.
>
> ***"CoT data undergoes Chinese character detection and missing coordinate detection" - why "Chinese character"?***
>
> We utilized QwenVL-Max to generate initial CoT, and it is mainly pre-trained with Chinese and English corpora. Therefore, Chinese characters may be generated in the output, which is a phenomenon we do not expect.
>
> Thanks again for the high-quality review. We hope our response could address your main concerns and we are happy to answer any further questions.

---

> > ### Comment · Reviewer_XCdi · 2025-08-04
> >
> > Thank you for your clarification.
> >
> > I would actually like to ask a question about the human study mentioned in response to reviewer 6t9d. Could you please provide more details on the experiment? E.g., which instruction the expert was given, did they get an ground truth samples, how much time they were given for each question, and are there any potential biases in the evaluations because of the writing style differences?

---

> > > ### Author Response · Authors · 2025-08-05
> > > **Official Comment by Authors**
> > >
> > > Thanks for the follow-up questions!
> > >
> > > Yes, I will provide you with more details of the experiment. We conducted the human expert experiment using all the test samples from the fine-tuning experiment. To fully reflect the researchers' capabilities and knowledge in industrial anomaly detection tasks and highlight the differences in learning patterns between humans and MLLMs before officially performing the tasks, we chose to only retain the test questions and options without setting additional prompts. At the same time, we ensured that the experts had not reviewed the corresponding ground truth samples before the test. In fact, we also built a simple  answer interface to avoid the experts making mistakes due to other factors during the test and to facilitate the statistics of the time consumed and the submitted answers after the test. Lastly, the accuracy rate of the expert test reached 62.67%, and the average time spent on each question was approximately 5 seconds. Please note that the experts  also generated thought chains during the thinking process of the test, but they were not explicitly expressed. For the potential biases that may exist in the experiment, we will conduct a thorough discussion in subsequent more comprehensive experiments.

---

> > > > ### Comment · Reviewer_XCdi · 2025-08-08
> > > >
> > > > Thank you for your further clarification.

---

### Comment · Area_Chair_ysCu · 2025-08-06
**Please engage in the reviewer-author discussion.**

Dear Reviewers,

Thank you for providing the initial reviews. Please respond to the authors' rebuttal and engage in the reviewer-author discussion if you haven't done so.

Best,

AC

---

### Decision · Program_Chairs · 2025-09-18

**Decision:**

Accept (poster)

**Comment:**

The reviewers agree that this paper makes a timely and impactful contribution by introducing AnomalyCoT, a multimodal Chain-of-Thought dataset for industrial anomaly detection (IAD). The dataset addresses critical gaps in existing resources by providing structured reasoning processes, diverse scenarios, and precise anomaly localization. The proposed data generation pipeline ensures accurate and interpretable CoT annotations, enabling unique evaluation of MLLMs’ reasoning in IAD. The authors conduct comprehensive experiments across commercial and open-source models, with results showing substantial performance gains, which highlights the dataset’s practical utility. Overall, the paper is well-executed, clearly demonstrates novelty, and offers a valuable benchmark for advancing multimodal reasoning in IAD.